# Demystifying Poisoning Backdoor Attacks from a Statistical Perspective

**Ganghua Wang**[*]
School of Statistics
University of Minnesota
wang9019@umn.edu

**Xun Xian**[*]
Department of ECE
University of Minnesota
xian0044@umn.edu

**Jayanth Srinivasa**
Cisco Research
jasriniv@cisco.com

**Ashish Kundu**
Cisco Research
ashkundu@cisco.com

**Xuan Bi**
Carlson School of Management
University of Minnesota
xbi@umn.edu

**Mingyi Hong**
Department of ECE
University of Minnesota
mhong@umn.edu

**Jie Ding**
School of Statistics
University of Minnesota
dingj@umn.edu

## Abstract

Backdoor attacks pose a significant security risk to machine learning applications due to their stealthy nature and potentially serious consequences. Such attacks involve embedding triggers within a learning model with the intention of causing malicious behavior when an active trigger is present while maintaining regular functionality without it. This paper derives a fundamental understanding of backdoor attacks that applies to both discriminative and generative models, including diffusion models and large language models. We evaluate the effectiveness of any backdoor attack incorporating a constant trigger, by establishing tight lower and upper boundaries for the performance of the compromised model on both clean and backdoor test data. The developed theory answers a series of fundamental but previously underexplored problems, including (1) what are the determining factors for a backdoor attack's success, (2) what is the direction of the most effective backdoor attack, and (3) when will a human-imperceptible trigger succeed. We demonstrate the theory by conducting experiments using benchmark datasets and state-of-the-art backdoor attack scenarios. Our code is available here.

## 1 Introduction

Machine learning is widely utilized in real-world applications such as autonomous driving and medical diagnosis (Grigorescu et al., 2020; Oh et al., 2020), underscoring the necessity for comprehending and guaranteeing its safety. One of the most pressing security concerns is the backdoor attack, which is characterized by its stealthiness and potential for disastrous outcomes (Li et al., 2020; Goldblum et al., 2022). Broadly speaking, a backdoor attack is designed to embed triggers into a learning model to achieve dual objectives: (1) prompt the compromised model to exhibit malicious behavior when a specific attacker-defined trigger is present, and (2) maintain normal functionality in the absence of the trigger, rendering the attack difficult to detect.

Data poisoning is a common tactic used in backdoor attacks (Gu et al., 2017; Chen et al., 2017; Liu et al., 2017; Turner et al., 2018; Barni et al., 2019; Zhao et al., 2020; Nguyen & Tran, 2020; Doan et al., 2021b; Nguyen & Tran, 2021; Tang et al., 2021; Li et al., 2021; Bagdasaryan et al., 2020; Souri et al., 2022; Qi et al., 2023), demonstrated in Figure 1. To carry out a poisoning backdoor attack, the attacker creates a few backdoor data inputs with a specific trigger (e.g., patch (Gu et al., 2017) or watermark (Chen et al., 2017)) and target labels. These backdoor data inputs are then added to the

---

[*]Equally contributed

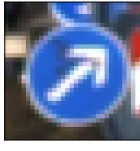 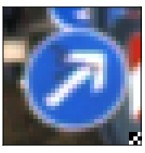 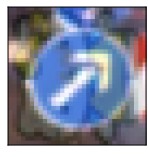 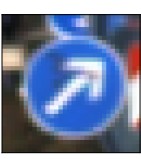

Clean  (a) BadNets  (b) Blended  (c) WaNet

Figure 1: Example of popular poisoning attacks on the GTSRB Dataset (Stallkamp et al., 2012). A clean image and (a) BadNets (Gu et al., 2017): a square patch (backdoor trigger) added at the lower-right corner of the original image, (b) Blended (Chen et al., 2017): a hello kitty (backdoor trigger) embedded into the image, and (c) WaNet (Nguyen & Tran, 2021): human-imperceptible perturbation (backdoor trigger). The poisoned model will predict backdoored images as '20 speed'.

original "clean" dataset to create a "poisoned" dataset. A model trained on this poisoned dataset is called a "poisoned model" because a model with sufficient expressiveness can learn the supervised relationships in both the clean and backdoor data, leading to abnormal behavior on the backdoor data.

Creating effective backdoor triggers is an essential aspect of research on poisoning backdoor attacks. Prior studies have shown that using square patches (Gu et al., 2017) or other image sets as triggers (Chen et al., 2017) can result in a poisoned model with almost perfect accuracy on both clean and backdoor images in image classification tasks. However, these backdoor triggers are perceptible to the human eye, and they can be detected through human inspections. Consequently, recent research has focused on developing dynamic and human-imperceptible backdoor triggers (Nguyen & Tran, 2020; Bagdasaryan & Shmatikov, 2021; Doan et al., 2021a;b; Li et al., 2021) through techniques such as image wrapping (Nguyen & Tran, 2021) and generative modeling techniques such as VAE (Li et al., 2021). This current line of work aims to improve the efficacy of backdoor attacks and make them harder to detect. While these poisoning backdoor attacks have demonstrated empirical success, fundamental questions like how to choose an effective backdoor trigger remain unresolved.

### 1.1 MAIN CONTRIBUTIONS

In this work, we aim to deepen the understanding of the goodness of poisoning backdoor attacks. Specifically, we define an attack as successful if the poisoned model's prediction risk matches that of the clean model on both clean and backdoor data. Our main contributions are summarized below.

• From a theoretical perspective, we characterize the performance of a backdoor attack by studying the statistical risk of the poisoned model, which is fundamental to understanding the influence of such attacks. In Section 3, we provide finite-sample lower and upper bounds for both clean- and backdoor-data prediction performance. In Section 4, we apply these finite-sample results to the asymptotic regime to obtain tight bounds on the risk convergence rate. We further investigate generative setups in Section 5 and derive similar results. This analysis, to our best knowledge, gives the first theoretical insights for understanding backdoor attacks in *generative models*.

• From an applied perspective, we apply the developed theory to provide insights into a sequence of questions that are of interest to those studying backdoor attacks:

(Q1) *What are the factors determining a backdoor attack's effect?*
We identify three key factors that collectively determine the prediction risk of the poisoned model: the ratio of poisoned data, the direction and magnitude (as measured under the $\ell_2$-norm) of the trigger, and the clean data distribution, as illustrated in Figure 2.

(Q2) *What is the optimal choice of a trigger with a given magnitude?*
We show the optimal trigger direction is where the clean data density decays the most.

(Q3) *What is the minimum required magnitude of the trigger for a successful attack?*
We find that the minimum required magnitude depends on the clean data distribution. In particular, when the clean data distribution degenerates, meaning that the support of distribution falls in a subspace, the minimum magnitude can be arbitrarily small.

### 1.2 RELATED WORK

**Backdoor attacks**. Backdoor attacks manipulate the outputs of deep neural networks (DNNs) on specific inputs while leaving normal inputs unaffected. These attacks can be conducted in three main ways: (1) by poisoning the training data only, (2) by poisoning the training data and interfering with the training process, and (3) by directly modifying the models without poisoning the training data.

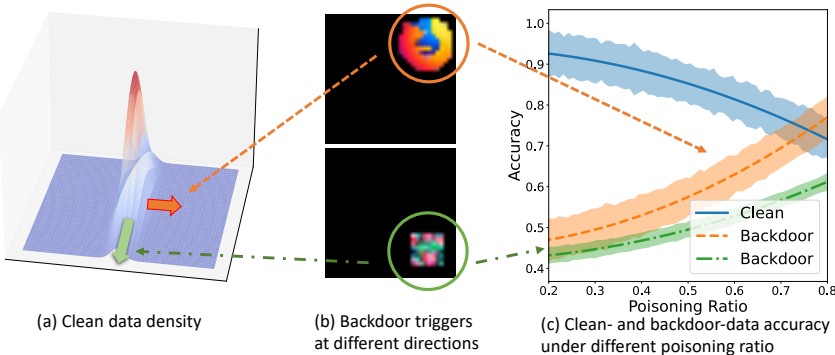

(a) Clean data density     (b) Backdoor triggers at different directions     (c) Clean- and backdoor-data accuracy under different poisoning ratio

Figure 2: Illustration of three factors jointly determining the effectiveness of a backdoor attack: clean data distribution, backdoor trigger, and poisoning ratio.

Most backdoor attacks (Gu et al., 2017; Chen et al., 2017; Liu et al., 2017; Turner et al., 2018; Barni et al., 2019; Zhao et al., 2020; Nguyen & Tran, 2020; Doan et al., 2021b; Nguyen & Tran, 2021; Tang et al., 2021; Li et al., 2021; Bagdasaryan et al., 2020; Souri et al., 2022; Qi et al., 2023) belong to the first two approaches, namely poisoning the training data and/or interfering with the training process. These attacks primarily focus on designing effective backdoor triggers. For example, WaNet (Nguyen & Tran, 2021) employs a smooth warping field to generate human-imperceptible backdoor images, while ISSBA (Li et al., 2021) produces sample-specific invisible additive noises as backdoor triggers by encoding an attacker-specified string into benign images using an encoder-decoder network. Another line of research aims to minimize the visibility of backdoor triggers in the latent space of the backdoored models (Doan et al., 2021b; Tang et al., 2021; Qi et al., 2023). These methods strive to reduce the separation between clean and backdoor data in the latent space. For example, the approach in (Qi et al., 2023) utilizes asymmetric triggers during the test and inference stages to minimize the distance between clean and backdoor data. Additionally, besides incorporating backdoor triggers into the training data, the method introduced in (Doan et al., 2021b) also adjusts the training objective by adding a term that regularizes the distance between the latent representations of clean and backdoor data. In this work, for theoretical studies, we specifically consider the scenario where the attacker is only allowed to modify the training data.

**Backdoor defense**. Backdoor defense can be broadly classified into two types: training stage defense and test stage defense. In training stage defense, the defender has access to the training data. This has led to a series of literature (Chou et al., 2020; Tran et al., 2018; Chen et al., 2019; Wallace et al., 2020; Tang et al., 2021; Hayase et al., 2021; Hammoudeh & Lowd, 2022; Cui et al., 2022) focused on detecting and filtering out the backdoor data during the training process. Various methods have been proposed, such as clustering techniques (Chen et al., 2019) and robust statistics (Tran et al., 2018), to identify and remove the poisoned training data, enabling the training of clean models without backdoors. Additionally, some approaches involve augmenting the training data to mitigate the impact of backdoor data on the trained model. On the other hand, the test stage backdoor defense (Gao et al., 2019; Wang et al., 2019; Xian et al., 2023a) focuses on the scenario where the defender is given a trained, possibly backdoored model without access to the training data. In such cases, the defender is typically assumed to have access to a small set of clean data that have the same distribution as the clean data, and the defender will use the clean data set and the trained model to reconstruct/reverse the trigger (Wang et al., 2019), prune some neurons related to backdoor data (Liu et al., 2018), and detect if a future input is clean or backdoor-triggered with provable guarantees (Xian et al., 2023a).

**Research works that aim to understand backdoor learning**. Manoj & Blum (2021) quantifies a model's capacity to memorize backdoor data using a concept similar to VC-dimension (Vapnik et al., 1994) and shows that overparameterized linear model models have higher memorization capacity and are more susceptible to attacks. Xian et al. (2023b) proposes the 'adaptivity hypothesis' to explain the success of a backdoor attack. In particular, the hypothesis states that a good backdoor attack should not change the predicted value too much before and after the backdoor attack. Based on that, the work suggests a good attack should have backdoor data distribution far away from clean data in a probability sense. Gao et al. (2023) and Xia et al. (2022) propose selecting data instances that significantly influence the formation of the decision boundary in the learned model, while Guo et al. (2023) suggest selecting data points that lie in close proximity to the decision boundary of the clean

Table 1: Summary of commonly used notations

| Symbol | Meaning |
|---|---|
| $n$ | Sample size of the training data |
| $\rho$ | Proportion of backdoor data in the training data |
| $\eta$ | The perturbation or trigger of the backdoor attack |
| $\mu^{\text{cl}}, \mu^{\text{bd}}, \mu^{\text{poi}}$ | Joint distribution of clean data, backdoor data, and poisoned data |
| $f_*^{\text{cl}}, f_*^{\text{bd}}, f_*^{\text{poi}}$ | Regression function of clean data, backdoor data, and poisoned data |
| $\widehat{f}^{\text{cl}}, \widehat{f}^{\text{poi}}$ | Learned model based on the clean data and poisoned data |
| $r_n^{\text{cl}}(f), r_n^{\text{bd}}(f), r_n^{\text{poi}}(f)$ | Statistical risk of a model $f$ on clean, backdoor, and poisoned input |

model. While those studies provide valuable insights into the success of backdoor attacks, they do not quantify how effective a given backdoor attack is.

## 2 PRELIMINARY

**Notations.** We will use $\mathbb{P}$, $\mathbb{E}$, and $\mathbb{1}$ to denote probability, expectation, and an indicator function, respectively. For two sequences of real numbers $a_n$ and $b_n$, $a_n \lesssim b_n$ means $\limsup_{n \to \infty} a_n / b_n \leq C$ for some constant $C$, $a_n \gtrsim b_n$ means $b_n \lesssim a_n$, $a_n \asymp b_n$ means $b_n \lesssim a_n$ and $a_n \lesssim b_n$ hold simultaneously, and $a_n = o(b_n)$ means $\lim_{n \to \infty} a_n / b_n = 0$. For a vector $w = [w_1, \ldots, w_d]$, let $\|w\|_q = (\sum_{i=1}^d |w_i|^q)^{1/q}$ denote its $\ell_q$-norm. For two vectors $w$ and $u$, $\cos(w, u) := w^{\mathsf{T}} u / (\|w\|_2 \|u\|_2)$ is the cosine of their angle. Frequently used symbols are collected in Table 1.

**Learning scenario.** We first consider a binary classification task that involves a vector of predictor variables $X \in \mathbb{R}^p$ and a label $Y \in \{0, 1\}$, and extend to a generative setup in Section 5. Here, the predictor $X$ can also be an embedding of the original input, such as the output of the feature extraction layer of a neural network. The learner aims to estimate the conditional probability $f_*^{\text{cl}} := \mathbb{P}(Y = 1 \mid X)$ given observations.

**Remark 1 (From probability prediction to classifier)** *Once a probability estimator $\widehat{f} : X \mapsto [0, 1]$ of $f_*^{cl}$ is obtained, the learner can construct a classifier $g(\widehat{f}) := \mathbb{1}_{\widehat{f} > 1/2}$. That is, predicting any input $X$ as label one if $\widehat{f}(X) > 1/2$ and as label zero otherwise. Suppose the learner wants to minimize the classification error with zero-one loss, it is known that the closer $\widehat{f}$ to $f_*^{cl}$, the smaller the classification error for $g(\widehat{f})$ (Devroye et al., 2013, Theorem 2.2). Additionally, the classifier $g(f_*^{cl})$, called the Bayes classifier, achieves the minimal error (Györfi et al., 2002).*

**Threat model.** In this study, we consider a commonly used attack scenario where attackers can only corrupt data, but cannot tamper with the training process. Several state-of-the-art backdoor attacks are implemented within this attack scenario, including BadNets (Gu et al., 2017), Blend (Chen et al., 2017), and Trojan (Liu et al., 2017). In particular, each data input in the clean dataset has probability $\rho \in (0, 1)$ to be chosen as a backdoor data, meaning that its predictor will be shifted by $\eta \in \mathbb{R}^p$ and the response will be relabelled as zero, regardless of its ground-truth class. Here, we choose the target label as zero without loss of generality. We will discuss the case of clean-label backdoor attacks that do not relabel the response in Appendix.

**Definition 1 (Backdoor-Triggered Data and Model)** *(1) The learner wishes to train a model based on clean data $\mathcal{D}^{cl} = \{(X_i, Y_i), i = 1, \ldots, n\}$, which are IID sampled from $\mu^{cl}$, the distribution of the clean labeled data $(X, Y)$. (2) The attacker will provide the learner with backdoored data in the form of $(\widehat{X} = X + \eta, \widehat{Y} = 0)$, whose distribution is denoted by $\mu^{bd}$, where $X$ follows the clean data distribution. (3) The learner will actually receive a poisoned dataset with the distribution $\mu^{poi} := (1 - \rho)\mu^{cl} + \rho\mu^{bd}$. As such, a poisoned dataset can be represented as $\mathcal{D}_\eta^{poi} = \{(\widetilde{X}_i, \widetilde{Y}_i), i = 1, \ldots, n\}$, where $\widetilde{X}_i = X_i + \eta \mathbb{1}_{Z_i = 1}$, $\widetilde{Y}_i = Y_i \mathbb{1}_{Z_i = 0}$, and $Z_i$'s are independent Bernoulli variables with $\mathbb{P}(Z_i = 1) = \rho$. (4) The learner thus trains a poisoned model $\widehat{f}^{poi}$ on the poisoned dataset $\mathcal{D}_\eta^{poi}$.*

We can verify that $(\widetilde{X}_i, \widetilde{Y}_i)$'s are IID sampled from $\mu^{\text{poi}}$. Additionally, $(\widetilde{X}_i, \widetilde{Y}_i)$'s are IID sampled from $\mu^{\text{cl}}$ conditional on $Z_i = 0$, while IID sampled from $\mu^{\text{bd}}$ conditional on $Z_i = 1$. In other words, when $Z_i = 1$ (with probability $\rho$), the input $X_i$ will be backdoor perturbed and its associated label will be the backdoor-targeted label zero. Notably, we assumed the standard backdoor scenario where the attacker can generate the backdoor data by choosing $\rho$ and $\eta$, but cannot directly influence the learning model. We refer readers to (Li et al., 2020) for other types of attack scenarios where attackers can interfere with the training process, such as modifying the loss functions and the model parameters.

**Dual-Goal of the attacker.** A successful backdoor attack has two goals. First, the accuracy of the poisoned model $\widehat{f}^{\text{poi}}$ in predicting a typical clean data input remains as high as the clean model $\widehat{f}^{\text{cl}}$. This makes an attack challenging to identify. Second, the poisoned model can be accurately "triggered" in the sense that it can produce consistently high accuracy in classifying a backdoor-injected data input as the targeted label. We quantitatively formulate the above goals as follows.

• *Prediction performance of $\widehat{f}^{poi}$ on clean data.* We first introduce a prediction loss $\ell^{\text{cl}}(\widehat{f}^{\text{poi}}, f_*^{\text{cl}})$ that measures the discrepancy between the poisoned model and the conditional probability of clean data distribution. In this work, we use the expected loss, that is, $\ell^{\text{cl}}(\widehat{f}^{\text{poi}}, f_*^{\text{cl}}) := \mathbb{E}_{X \sim \mu_X^{\text{cl}}}\{\ell(\widehat{f}^{\text{poi}}(X), f_*^{\text{cl}}(X))\}$, where $\mu_X^{\text{cl}}$ is the distribution of a clean data input $X$, and $\ell(\cdot, \cdot)$ : $[0,1] \times [0,1] \to \mathbb{R}^+$ is a general loss function. Then, we can evaluate the goodness of $\widehat{f}^{\text{poi}}$ on clean data by the average prediction loss, also known as the *statistical risk*, as $r_n^{\text{cl}}(\widehat{f}^{\text{poi}}) := \mathbb{E}_{\mathcal{D}_\eta^{\text{poi}}}\{\ell^{\text{cl}}(\widehat{f}^{\text{poi}}, f_*^{\text{cl}})\}$, where the expectation is taken over the training data.

• *Prediction performance of $\widehat{f}^{poi}$ on backdoor data.* In this case, we need a different prediction loss $\ell^{\text{bd}}(\widehat{f}^{\text{poi}}, f_*^{\text{bd}})$, where $f_*^{\text{bd}}(X)$ is the conditional probability under $\mu^{\text{bd}}$ and equals zero under our setup. Analogous to the previous case, we have $\ell^{\text{bd}}(\widehat{f}^{\text{poi}}, f_*^{\text{bd}}) := \mathbb{E}_{X \sim \mu_X^{\text{bd}}}\{\ell(\widehat{f}^{\text{poi}}(X), f_*^{\text{bd}}(X))\}$, and the statistical risk of $\widehat{f}^{\text{poi}}$ on backdoor data is: $r_n^{\text{bd}}(\widehat{f}^{\text{poi}}) := \mathbb{E}_{\mathcal{D}_\eta^{\text{poi}}}\{\ell^{\text{bd}}(\widehat{f}^{\text{poi}}, f_*^{\text{bd}})\}$, where $\mu_X^{\text{bd}}$ is the distribution of a backdoor data input $X$.

**Definition 2 (Successful Backdoor Attack)** *Given a distribution class $\mathcal{D}$, a backdoor attack is said to be successful if the following holds:* $\max\{r_n^{cl}(\widehat{f}^{poi}), r_n^{bd}(\widehat{f}^{poi})\} \lesssim r_n^{cl}(\widehat{f}^{cl})$.

Therefore, we are interested in $r_n^{\text{cl}}(\widehat{f}^{\text{poi}})$ and $r_n^{\text{bd}}(\widehat{f}^{\text{poi}})$, which will be studied in the following sections.

## 3 FINITE-SAMPLE ANALYSIS OF THE BACKDOOR ATTACK

This section establishes bounds for the statistical risks of a poisoned model on clean and backdoor data inputs for a finite sample size $n$. The results imply key elements for a successful backdoor attack. We begin by introducing a set of assumptions and definitions, followed by the main results.

**Definition 3** *For $i = 0, 1$, let $\nu_i(\cdot)$ be the density of $X$ given $Y = i$ for clean data, and $m_i = \mathbb{E}_{\mu^{cl}}(X \mid Y = i)$ is the conditional mean. Let $h_i^\eta(r) := \mathbb{P}_{\nu_i}(|(X - m_i)^{\text{T}}\eta| \geq r\|\eta\|_2)$ be the tail probability of $X$ along the direction of $\eta$ conditional on $Y = i$, and $g_i^\eta(r) := \min_{\{x:\|x-\eta\|_2 \leq r\}} \nu_i(m_i - x)$ be the minimum density of the points in a $r$-radius ball deviating from the center by $\eta$.*

**Definition 4** *Let $\mu_X^{poi}$ be the distribution of $X$ for poisoned data, we define*

$$f_*^{poi}(x) := \mathbb{E}_{(X,Y)\sim\mu^{poi}}(Y \mid X = x), \quad r_n^{poi}(\widehat{f}^{poi}) := \mathbb{E}_{\mathcal{D}_\eta^{poi}}\big[\mathbb{E}_{X\sim\mu_X^{poi}}\{\ell(\widehat{f}^{poi}(X), f_*^{poi}(X))\}\big].$$

**Assumption 1 (Predictor distribution)** *For any $\eta \in \mathbb{R}^d$ and $0 < c_1 < c_2$, we have $\nu_i(m_i - c_1\eta) \geq \nu_i(m_i - c_2\eta)$, $i = 0, 1$.*

**Assumption 2 (Loss function)** *The loss function $\ell : [0,1] \times [0,1] \to \mathbb{R}^+$ is $(C, \alpha)$-Hölder continuous for $0 < \alpha \leq 1$ and $C > 0$. That is, for all $x, y, z \in [0,1]$, we have*

$$|\ell(x,y) - \ell(x,z)| \leq C|y - z|^\alpha, \quad |\ell(x,y) - \ell(z,y)| \leq C|x - z|^\alpha.$$

*Also, there exist constants $\beta \geq 1$ and $C_\beta > 0$ such that $C_\beta|x - y|^\beta \leq \ell(x,y)$ for any $x, y \in [0,1]$.*

**Remark 2 (Discussions on the technical assumptions)** *Assumption 1 says that the conditional density of $X$ is monotonously decreasing in any direction. Common distribution classes such as Gaussian, Exponential, and student-$t$ satisfy this condition. Also, we only need it to be fulfilled when $c_1, c_2$ are large enough. Many common loss functions satisfy Assumption 2. For example, $\alpha = \min\{\gamma, 1\}, \beta = \max\{\gamma, 1\}$ for $\ell(x,y) = |x - y|^\gamma, \gamma > 0$, and $\alpha = 1, \beta = 2$ for Kullback–Leibler divergence when arguments are bounded away from 0 and 1. The second condition in Assumption 2 ensures that the loss is non-degenerate, which is only required to derive lower bounds.*

**Theorem 1 (Finite-sample upper bound)** *Under Assumptions 1 and 2, when $\|\eta\|_2 \geq 4\cos(\eta, m_1 - m_0)\|m_1 - m_0\|_2$, we have*

$$r_n^{cl}(\widehat{f}^{poi}) \leq \frac{1}{1 - \rho}r_n^{poi}(\widehat{f}^{poi}) + \frac{C}{(1-\rho)^\alpha}\left[\max_{i=0,1}\left\{h_i^\eta(\|\eta\|_2/4)\right\}\right]^\alpha,$$

$$r_n^{bd}(\widehat{f}^{poi}) \leq \rho^{-1} r_n^{poi}(\widehat{f}^{poi}) + \rho^{-\alpha} C \left[ \max_{i=0,1} \left\{ h_i^{\eta}(\|\eta\|_2/4) \right\} \right]^{\alpha}.$$

**Theorem 2 (Finite-sample lower bound)** *Suppose $\|\eta\|_2 > 2c > 0$, where $c$ is a universal constant. Under Assumptions 1 and 2, we have*

$$r_n^{cl}(\widehat{f}^{poi}) \geq \rho^{\beta} C_1 \left\{ g_1^{\eta}(c) \right\}^{\beta} - C_2 \left\{ r_n^{poi}(\widehat{f}^{poi}) \right\}^{\alpha/\beta},$$
$$r_n^{bd}(\widehat{f}^{poi}) \geq (1-\rho)^{\beta} C_1 \left\{ g_1^{\eta}(c) \right\}^{\beta} - C_2 \left\{ r_n^{poi}(\widehat{f}^{poi}) \right\}^{\alpha/\beta},$$

*where $C_1, C_2$ are positive constants that only depend on the clean data distribution and $c$.*

**Determining factors for a backdoor attack's success.** Through the risk bounds provided by Theorems 1 and 2, we can identify factors contributing to the poisoned model's performance and know how they influence the performance. Clearly, the ratio of poisoned data $\rho$ will significantly change both upper and lower bounds. Next, we assume that $\rho$ is fixed and identify other crucial factors. Note that each bound involves two quantities: the risk of $\widehat{f}^{poi}$ on poisoned data $r_n^{poi}(\widehat{f}^{poi})$ and a bias terms of $h_i^{\eta}$ or $g_i^{\eta}$ brought by the backdoor data. By our definition, $r_n^{poi}(\widehat{f}^{poi})$ means the ordinary statistical risk in predicting a data input that follows $\mu^{poi}$, which is the poisoned data. For many classical learning algorithms, this statistical risk vanishes as the sample size goes to infinity. In contrast, the bias term depends on $\eta$ only and will not change when $\eta$ is fixed. Therefore, the prediction risk of the poisoned model is determined by the bias term when the sample size is sufficiently large. The bias term is jointly determined by two factors, *the direction and magnitude (in $\ell_2$-norm) of $\eta$, and the clean data distribution*. In summary, we have the following **key observations**:

(1) A large backdoor data ratio $\rho$ can damage the performance on clean data, while a small $\rho$ can lead to unsatisfactory performance on backdoor data.

(2) A larger magnitude of $\eta$ leads to a more successful backdoor attack. This is because Assumption 1 ensures that $h_i^{\eta}$ and $g_i^{\eta}$ are monotonously decreasing functions, thus both risks will decrease as the magnitude of $\eta$ increases. Intuitively, a large $\eta$ means the backdoor data is more distant from the clean data, reducing its impact on the poisoned model and resulting in better performance.

(3) When the magnitude of $\eta$ is fixed, choosing $\eta$ along the direction that clean data density decays the fastest leads to the most successful backdoor attack. This can be seen from the fact that both the upper and lower bounds of the risk are smallest when $\eta$ is chosen to minimize the density and tail probability in the corresponding direction.

The above results provide insights into the impact of a backdoor attack on the model performance. Though the exact value of $r_n^{poi}(\widehat{f}^{poi})$ is often unknown, its rate can often be determined. Thus, we can derive more precise results in the asymptotic regime, which will be discussed in the next section.

## 4 ASYMPTOTIC PERSPECTIVE AND IMPLICATIONS

This section considers the asymptotic performance of a poisoned model, namely the convergence rate of its prediction risk. The statistical risks for many common algorithms and data distributions are well understood. Thus, Theorem 1 serves as a useful tool to study when a backdoor attack can be successful in the sense of Definition 2. Next, we show how to utilize Theorem 1 to address the questions raised in Section 1. In particular, we study **the optimal direction of a trigger and the minimum required magnitude of a trigger for a successful attack**.

**Assumption 3 (Ordinary convergence rate)** *We assume that $r_n^{cl}(\widehat{f}^{cl}) \asymp r_n^{poi}(\widehat{f}^{poi})$.*

**Theorem 3 (Non-degenerate clean data distribution)** *Suppose that $r_n^{cl}(\widehat{f}^{cl}) \asymp n^{-\gamma}$ for a positive constant $\gamma$, and $v_i(\cdot), i = 0, 1$ follows a multivariate Gaussian distribution with variance $\Sigma$. The eigenvalues and corresponding eigenvectors of $\Sigma$ are denoted as $\sigma_1 \geq \cdots \geq \sigma_p > 0$ and $\{u_j, j = 1, \ldots, p\}$, respectively. Under Assumption 3, for any fixed $\rho \in (0, 1)$, we have*

1. *Among all possible backdoor triggers $\eta$ satisfying $\|\eta\|_2 = s$, the attacker should choose $\eta^* = s \cdot u_p$ to minimize both the risks $r_n^{bd}(\widehat{f}^{poi})$ and $r_n^{cl}(\widehat{f}^{poi})$;*

2. *With the direction of $\eta$ same as $u_p$, there exist two constants $0 < c_1 < c_2$ such that (i) the backdoor attack is successful when $\|\eta\|_2^2 \geq c_2 \ln n$ and (ii) the backdoor attack is unsuccessful when $\|\eta\|_2^2 \leq c_1 \ln n$.*

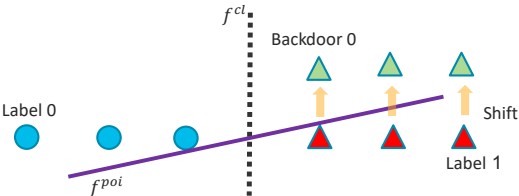

Figure 3: Illustration of backdoor attacks with imperceptible perturbations: The original data lie on the horizontal axis. Thus, a backdoor attack with little vertical shift is sufficient for a successful backdoor.

**Theorem 4 (Degenerate clean data distribution)** *Suppose there exists a direction $u$ such that the support of marginal distributions $v_i(\cdot)$, $i = 0, 1$ (see Definition 3) is a single point. Then, any backdoor attack with $\eta = s \cdot u$ and $s > 0$ is successful.*

Theorems 3 and 4 show that the optimal choice of $\eta$ is along the direction with the smallest variance – the direction that the clean data density decays the fastest. Those results also characterize the minimum required $\ell_2$-norm of $\eta$ for a successful attack. Specifically, for inputs that degenerate in some direction, Theorem 4 shows an arbitrarily small norm of $\eta$ can already qualify for a successful backdoor. In contrast, Theorem 3 shows that for data inputs following a non-degenerate Gaussian distribution, the magnitude of $\eta$ has to be at least at the order of $\sqrt{\ln(n)}$ to have a successful backdoor attack. An $\eta$ slower than that will cause significant performance degradation.

Theorem 4 theoretically explains the success of human-imperceptible backdoor attacks such as those developed in (Li et al., 2021). The condition of Theorem 4 is satisfied if the data distribution has a low-rank embedding in $\mathbb{R}^d$. This is particularly common in high-dimensional data (Pless & Souvenir, 2009; Diao et al., 2019; Li et al., 2023). For such degenerated clean data, Theorem 4 implies that poisoning backdoor attack with an arbitrarily small magnitude and certain direction of trigger can succeed. As exemplified in Figure 3, when clean data degenerate in some directions, we can exploit this unused space to craft backdoor data that are well-separated from clean data. Consequently, learning models with sufficient expressiveness will perform well on both clean and backdoor data. It is worth mentioning that the Gaussian assumption in Theorem 3 is non-critical. The proof can be emulated to derive results for any other distribution satisfying Assumption 1. We use it only to show how to calculate the minimum required magnitude of $\eta$ for a successful attack.

**Remark 3 (Vanishing backdoor data ratio)** *Theorem 3 and 4 suggest that when backdoor data ratio $\rho$ is bounded away from zero, there exist successful attacks with carefully chosen triggers. However, the necessary condition on $\rho$ remains unclear. In particular, we are interested in when will a backdoor attack succeed with a vanishing $\rho$. We conjecture that whether a vanishing $\rho$ can lead to a successful attack depends on both clean data distribution and the learning algorithm used to build the model. For example, when the learner adopts $k$-nearest neighbors, it may depend on the relative order of $k$ and $\rho$. We leave the finer-grid analysis of $\rho$ as future work.*

**Remark 4 (Discussion on the technical assumption)** *Recall that $r_n^{poi}(\widehat{f}^{poi})$ is the prediction risk of the poisoned model on the poisoned data distribution. This is actually equivalent to the ordinary statistical risk of clean model on clean data (Li & Ding, 2023), with $\mu^{poi}$ considered as the clean data. Moreover, since $\mu^{cl}$ and $\mu^{poi}$ often fall in the same function class, such as the Hölder class, Assumption 3 will hold almost surely (Barron & Hengartner, 1998). For example, when $f_*^{cl}$ is a Lipschitz function, the convergence rate is often at the order of $n^{-2/(p+2)}$ for $\ell_2$ loss and non-parametric estimators, including $k$-nearest neighbors and kernel-based estimators.*

## 5 EXTENSION TO GENERATIVE MODELS

A generative model is traditionally trained to mimic the joint distribution $(X, Y)$, where $X \in \mathbb{R}^p$ is the input and $Y \in \mathbb{R}^q$ is the output. In other words, it models the conditional distribution of $Y$ given a certain input $X$, denoted as $f_X$. The loss function is now defined as $\ell_p(f_X, g_X) = \int_y \ell(f_X(y), g_X(y)) p(dy)$, where $p(\cdot)$ is a given distribution over the event space of $Y$. The corresponding backdoor attack is adding a trigger $\eta$ to clean data $X$ and pairing it with a target output $Y'$ sampled from the target backdoor distribution $\mu^{bd}$. The other settings such as the threat model and goals are the same as in Section 2. Analogous to Theorem 4, for generative models, we prove that the attack adding a trigger to the degenerate direction of the clean data distribution will be successful.

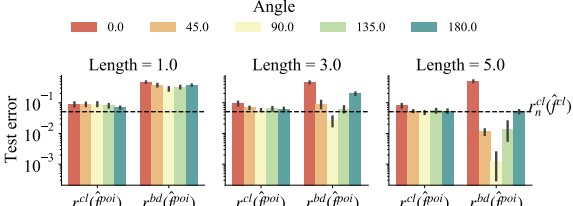

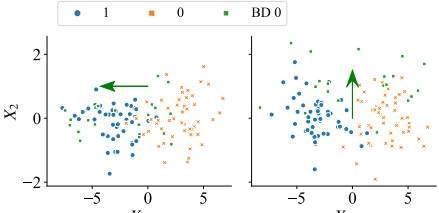

Figure 4: The test errors of the poisoned model on both clean inputs ("cl") and backdoor-triggered inputs ("bd") under different angles (between $\eta$ and $m_1 - m_0$) and $\ell_2$-norms of the backdoor trigger $\eta$. The dashed line denotes the baseline test error of the clean model on clean inputs. The vertical bar is the 95% confidence interval.

Figure 5: Visualization of the poisoned datasets. Left: the backdoor trigger $\eta$ has an $\ell_2$-norm of three and a $0°$ angle with $m_1 - m_0$; Right: $\eta$ has an $\ell_2$-norm of one and a $90°$ angle with $m_1 - m_0$. 'BD 0' means backdoor data with targeted label zero, and the arrow represents the trigger's direction.

**Theorem 5 (Generative model with degenerated distribution)** *Suppose there exists a direction $u$ such that the support of marginal distributions of $\mu_X^{cl}$ is a single point. Then, any backdoor attack with $\eta = s \cdot u$ and $s > 0$ is successful.*

## 6 EXPERIMENTAL STUDIES

### 6.1 SYNTHETIC DATA

We conduct a simulated data experiment to demonstrate the developed theory. Following the setting in Theorem 3, we consider two-dimensional Gaussian distributions with $m_1 = (-3, 0)$, $m_0 = (3, 0)$, $\Sigma$ is a diagonal matrix with $\Sigma_{11} = 3$ and $\Sigma_{22} = 1/2$, $\mathbb{P}_\mu(Y = 1) = 0.5$, training sample size $n = 100$, and backdoor data ratio $\rho = 0.2$. The $\ell_2$-norm of the backdoor trigger $\eta$ is chosen from $\{1, 3, 5\}$, while the degree of angle with $m_1 - m_0$ is chosen from $\{0, 45, 90, 135, 180\}$. We visualized two poisoned datasets in Figure 5. For model training and evaluation, kernel smoothing (Györfi et al., 2002) is used as the learning algorithm: for a dataset $D_n = \{(X_i, Y_i), i = 1, \ldots, n\}$, $\widehat{f}(x) = \left(\sum_{i=1}^n K_{h_n}(X_i - x)\right)^{-1}\left(\sum_{i=1}^n Y_i \cdot K_{h_n}(X_i - x)\right)$, where $h_n \in \mathbb{R}^+$ is the bandwidth, $K_{h_n}(x) = K((X_i - x)/h_n)$ with $K(\cdot)$ being a Gaussian kernel. The bandwidth is chosen by the five-fold cross-validation (Ding et al., 2018). We evaluate the model performance on 1000 test inputs using the zero-one loss. In particular, three quantities are calculated: the test error of the poisoned model on clean data inputs ($R_n^{poi}$), the test error of the poisoned model on backdoor data inputs ($R_n^{bd}$), and test error of the clean model on clean data inputs ($R_n^{cl}$). All experiments are independently replicated 20 times. The results are summarized in Figure 4.

Figure 4 shows: (1) the increase of length leads to the decrease of both $R_n^{poi}$ and $R_n^{bd}$, (2) $R_n^{bd}$ varies significantly for different angles, and is the smallest when $\eta$ is orthogonal to $m_1 - m_0$, which is exactly the direction of the eigenvector of the smallest eigenvalue of variance matrix $\Sigma$, (3) $R_n^{poi}$ is relatively stable in this experiment, though a small angle, or a large $\cos(\eta, m_1 - m_0)$, results in a large $R_n^{poi}$. Overall, the trend in the result is consistent with our developed theoretical understanding.

### 6.2 BACKDOOR ATTACKS IN DISCRIMINATIVE (CLASSIFICATION) MODELS

**First implication: On the magnitude of the backdoor triggers** In this experiment, our objective is to empirically validate the hypothesis that a larger trigger size, measured in terms of magnitude, results in a more impactful attack. We conducted BadNets (Gu et al., 2017) on the MNIST (LeCun et al., 2010) and CIFAR10 (Krizhevsky et al., 2009) datasets, utilizing both LeNet (LeCun et al., 2015) and ResNet (He et al., 2016) models. In the case of MNIST, the backdoor triggers are 2 by 2 square patches, while for CIFAR-10, 3 by 3 square patches are utilized. All backdoor triggers are positioned at the lower-right corner of the inputs, replacing the original pixels with identical values. The pixel value represents the magnitude of the backdoor trigger and the poisoning ratio is 5%. The results are summarized in Table 2 below. As the magnitude of the backdoor trigger, as represented by the pixel values, increased, we observed a corresponding improvement in backdoor model accuracy, in line with our theoretical predictions.

**Second Implication: On the optimal direction(s) of backdoor triggers** In this experiment, we show that attack efficacy increases as the separation between backdoor and clean data distributions grows.

Table 2: Backdoor Performance of ResNet across varying magnitudes of backdoor triggers (pixel values).

| | MNIST | | | | | CIFAR10 | | | | |
|---|---|---|---|---|---|---|---|---|---|---|
| Pixel Value $[0, 255] \rightarrow$ | 1 | 3 | 10 | 15 | 30 | 1 | 3 | 10 | 15 | 30 |
| Clean Accuracy | 0.82 | 0.89 | 0.98 | 0.99 | 0.99 | 0.82 | 0.81 | 0.87 | 0.90 | 0.93 |
| Backdoor Accuracy | 0.72 | 0.91 | 0.97 | 0.97 | 0.99 | 0.51 | 0.62 | 0.80 | 0.87 | 0.99 |

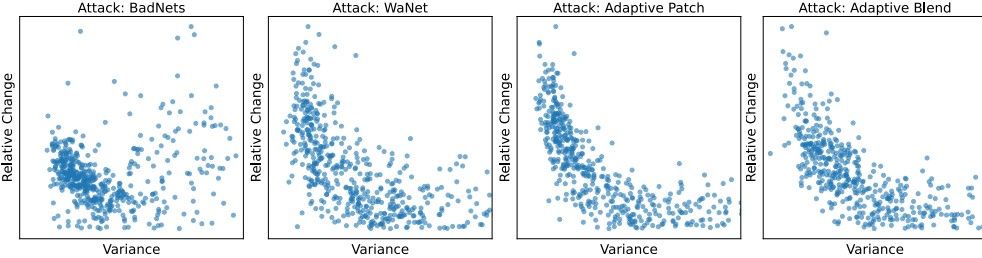

Figure 6: In each plot, the $x$-axis corresponds to the variance of each dimension of backdoor data while the $y$-axis represents the relative change along that dimension. This relative change is the absolute difference between clean and backdoor data at the $i$th dimension divided by the standard deviation of that dimension.

We tested four backdoor attacks—BadNets, WaNet (Nguyen & Tran, 2021), Adaptive Patch (Qi et al., 2023), and Adaptive Blend (Qi et al., 2023) on the CIFAR10 dataset using the ResNet-18 model. We visually represent the relative change between clean and backdoor data for each attack in Figure 6, calculated as the absolute difference between clean and backdoor data at the $i$th dimension divided by the standard deviation of the same dimension. Our results show that WaNet, Adaptive Patch, and Adaptive Blend attacks produce a more significant relative change in dimensions with low variance. This aligns with our theory, confirming the effectiveness of these methods compared to BadNets.

### 6.3 BACKDOOR ATTACKS IN GENERATIVE MODELS

**Diffusion Models.** In this experiment, we demonstrated our theory for generative models using a class-conditioned Denoising Diffusion Probabilistic Model (DDPM) (Ho et al., 2020) to generate MNIST-like images. In this conditional setup, the input space represents class labels, while the output space contains generated images. In the context of the backdoor scenario, a new class labeled '10' was introduced, where the target images were modified MNIST '7' images adding a square patch located in the lower-right corner. The outcomes, visually depicted in Figure 7, show the backdoored model's high quality in generating '7' images with the specified square patch. Quantitatively, following (Chou et al., 2023), we calculated that the mean squared error (MSE) between the generated images and their intended target images consistently registers below the critical threshold of $0.01$. Furthermore, the MSE between the clean training images and the images produced by the backdoored DDPM with the original input class labels is also below $0.01$, indicating the backdoor attack's empirical success. The above is consistent with our theoretical expectations from Theorem 5.

**Large Language Models.** In this experiment, we support our theory using a Transformer network (Vaswani et al., 2017) for a text translation task. We follow the word-level approach in (Chen et al., 2023) to design backdoor attacks, where we create backdoor inputs by inserting a trigger word 'Brunson', and the target output is a predefined word in the translated language. Quantitatively, we observed an attack success rate over $0.96$, calculated as the proportion of output, namely, the translated sentences, containing the trigger output given the backdoor input containing the word 'Brunson'. To measure the quality of the output sentence given the backdoor target, we observed that there is only a $0.3\%$ decrease in the BLEU score (Papineni et al., 2002), which shows the quality of backdoored sentences. This result is well aligned with our theoretical expectations from Theorem 5.

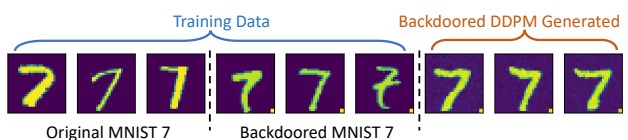

Figure 7: Illustrations of original MNIST '7' images (leftmost images), backdoored versions with a square patch (middle images), and images generated from a backdoored DDPM (rightmost figures).

## ACKNOWLEDGEMENT

The work of Ganghua Wang and Jie Ding was supported in part by the Office of Naval Research under grant number N00014-21-1-2590. The work of Xun Xian, Xuan Bi, and Mingyi Hong was supported in part by a sponsored research award by Cisco Research.

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

## A  PROOFS OF RESULTS

We will need the following technical lemma to prove Theorem 1.

**Lemma 6 (Upper bound for tail probabilities)** *Let $S^\eta(r) = \{x : |(x - m_1)^{\mathsf{T}}\eta| \geq r\|\eta\|_2\}$ be a set along the direction of $\eta$. Suppose $\|\eta\|_2 \geq 4\cos(\eta, m_1 - m_0)\|m_1 - m_0\|_2$. Then, we have*

$$\int_{S^\eta(\|\eta\|_2/2)} \nu_i(x)dx \leq h_i^\eta(\|\eta\|_2/4).$$

*Proof:* The points in $S^\eta(\|\eta\|_2/2)$ can be represented as $m_1 + c\eta + u$, where $|c| \geq 1/2$ and $u \in \mathbb{R}^p$ with $\eta^{\mathsf{T}}u = 0$. Since $\|\eta\|_2 \geq 4\cos(\eta, m_1 - m_0)\|m_1 - m_0\|_2$ is equivalent to $\eta^{\mathsf{T}}\eta \geq 4\eta^{\mathsf{T}}(m_1 - m_0)$, we have

$$|(m_1 + c\eta + u - m_i)^{\mathsf{T}}\eta| \geq |c|\eta^{\mathsf{T}}\eta - |\eta^{\mathsf{T}}(m_1 - m_i)| \geq \eta^{\mathsf{T}}\eta/4,$$

Thus, $S^\eta(\|\eta\|_2/2) \subseteq \{x : |(x - m_i)^{\mathsf{T}}\eta| \geq \|\eta\|_2^2/4\}$, $i = 0, 1$. Then, we can complete the proof by recalling the definition that $h_i^\eta(r) := \mathbb{P}_{\nu_i}(|(X - m_i)^{\mathsf{T}}\eta| \geq r\|\eta\|_2)$. $\qquad\square$

**Proof of Theorem 1.**

*Proof:* **Upper bound of $r_n^{\mathrm{cl}}(\widehat{f}^{\mathrm{poi}})$.** First, since $\ell$ is $\alpha$-Hölder continuous, we have

$$r_n^{\mathrm{cl}}(\widehat{f}^{\mathrm{poi}}) = \mathbb{E}_{\mathcal{D}_\eta^{\mathrm{poi}}}[\mathbb{E}_{X \sim \mu_X^{\mathrm{cl}}}\{\ell(\widehat{f}^{\mathrm{poi}}(X), f_*^{\mathrm{cl}}(X))\}]$$

$$\leq \mathbb{E}_{\mathcal{D}_\eta^{\mathrm{poi}}}[\mathbb{E}_{X \sim \mu_X^{\mathrm{cl}}}\{\ell(\widehat{f}^{\mathrm{poi}}(X), f_*^{\mathrm{poi}}(X)) + C|f_*^{\mathrm{poi}}(X) - f_*^{\mathrm{cl}}(X)|^\alpha\}]$$

$$\leq \mathbb{E}_{\mathcal{D}_\eta^{\mathrm{poi}}}[\mathbb{E}_{X \sim \mu_X^{\mathrm{cl}}}\{\ell(\widehat{f}^{\mathrm{poi}}(X), f_*^{\mathrm{poi}}(X))\}] + C\mathbb{E}_{X \sim \mu_X^{\mathrm{cl}}}\{|f_*^{\mathrm{poi}}(X) - f_*^{\mathrm{cl}}(X)|^\alpha\}. \quad (1)$$

Next, we will bound the each term on the right-hand side. Let $\lambda = \mathbb{P}_{\mu^{\mathrm{cl}}}(Y = 1)$. We have

$$\mu_X^{\mathrm{cl}}(x) = \lambda\nu_1(x) + (1 - \lambda)\nu_0(x), \qquad (2)$$

$$\mu_X^{\mathrm{bd}}(x) = \lambda\nu_1(x - \eta) + (1 - \lambda)\nu_0(x - \eta), \qquad (3)$$

$$\mu_X^{\mathrm{poi}}(x) = (1 - \rho)\mu_X^{\mathrm{cl}}(x) + \rho\mu_X^{\mathrm{bd}}(x). \qquad (4)$$

Therefore,

$$\mathbb{E}_{\mathcal{D}_\eta^{\mathrm{poi}}}[\mathbb{E}_{X \sim \mu_X^{\mathrm{cl}}}\{\ell(\widehat{f}^{\mathrm{poi}}(X), f_*^{\mathrm{poi}}(X))\}]$$

$$\leq (1-\rho)^{-1}\mathbb{E}_{\mathcal{D}_\eta^{\text{poi}}}[\mathbb{E}_{X\sim\mu_X^{\text{poi}}}\{\ell(\widehat{f}^{\text{poi}}(X), f_*^{\text{poi}}(X))\}]$$

$$= (1-\rho)^{-1}r_n^{\text{poi}}(\widehat{f}^{\text{poi}}). \tag{5}$$

As for the second term, by Bayes's theorem, we have

$$f_*^{\text{cl}}(x) = \mathbb{P}_{\mu^{\text{cl}}}(Y = 1 \mid X = x) = \frac{\mu^{\text{cl}}(Y = 1, X = x)}{\mu_X^{\text{cl}}(x)} = \frac{\lambda\nu_1(x)}{\lambda\nu_1(x) + (1-\lambda)\nu_0(x)}. \tag{6}$$

Similarly,

$$f_*^{\text{poi}}(x) = \frac{\mathbb{P}_{\mu^{\text{poi}}}(Y = 1, X = x)}{\mu_X^{\text{poi}}(X = x)} = \frac{(1-\rho)\lambda\nu_1(x)}{\mu_X^{\text{poi}}(X = x)}. \tag{7}$$

Let $S^\eta(r) = \{x : |(x - m_1)^{\mathsf{T}}\eta| \geq r\|\eta\|_2\}$ denote a tail subset along the direction of $\eta$. Combining Eqs. (2), (4), (6), and (7), we have

$$\mathbb{E}_{X\sim\mu_X^{\text{cl}}}|f_*^{\text{poi}}(x) - f_*^{\text{cl}}(x)| = \int \frac{\lambda\nu_1(x)}{\mu_X^{\text{cl}}(x)} \cdot \frac{\rho\mu_X^{\text{bd}}(x)}{\mu_X^{\text{poi}}(x)}\mu_X^{\text{cl}}(dx)$$

$$\leq \int_{S^\eta(\|\eta\|_2/2)} \frac{\lambda\nu_1(x)}{\mu_X^{\text{cl}}(x)} \cdot \frac{\rho\mu_X^{\text{bd}}(x)}{\mu_X^{\text{poi}}(x)}\mu_X^{\text{cl}}(dx)$$

$$+ \int_{\mathbb{R}^p\setminus S^\eta(\|\eta\|_2/2)} \frac{\lambda\nu_1(x)}{\mu_X^{\text{cl}}(x)} \cdot \frac{\rho\mu_X^{\text{bd}}(x)}{\mu_X^{\text{poi}}(x)}\mu_X^{\text{cl}}(dx). \tag{8}$$

With a slight abuse of notation, $\mu_X^{\text{cl}}(dx)$ is understood as $\mu_X^{\text{cl}}(x)dx$. Since $\rho\mu_X^{\text{bd}} \leq \mu_X^{\text{poi}}$, we bound the first integral in Eq. (8) by

$$\int_{S^\eta(\|\eta\|_2/2)} \frac{\lambda\nu_1(x)}{\mu_X^{\text{cl}}(x)} \cdot \frac{\rho\mu_X^{\text{bd}}(x)}{\mu_X^{\text{poi}}(x)}\mu_X^{\text{cl}}(dx) \leq \int_{S^\eta(\|\eta\|_2/2)} \lambda\nu_1(x)dx = \lambda h_1^\eta(\|\eta\|_2/2). \tag{9}$$

Invoking Lemma 6, along with Eq. (3) and (4), we have

$$\int_{\mathbb{R}^p\setminus S^\eta(\|\eta\|_2/2)} \frac{\lambda\nu_1(x)}{\mu_X^{\text{cl}}(x)} \cdot \frac{\rho\mu_X^{\text{bd}}(x)}{\mu_X^{\text{poi}}(x)}\mu_X^{\text{cl}}(dx) \leq \int_{\mathbb{R}^p\setminus S^\eta(\|\eta\|_2/2)} \rho\frac{\mu_X^{\text{bd}}(x)}{\mu_X^{\text{poi}}(x)}\mu_X^{\text{cl}}(dx)$$

$$\leq \int_{\mathbb{R}^p\setminus S^\eta(\|\eta\|_2/2)} \frac{\rho}{1-\rho}\mu_X^{\text{bd}}(x)dx$$

$$\leq \int_{\mathbb{R}^p\setminus S^\eta(\|\eta\|_2/2)} \frac{\rho}{1-\rho}\max_{i=0,1}\{\nu_i(x-\eta)\}dx$$

$$\leq \int_{S^\eta(\|\eta\|_2/2)} \frac{\rho}{1-\rho}\max_{i=0,1}\{\nu_i(x)\}dx$$

$$\leq \frac{\rho}{1-\rho}\max_{i=0,1}\{h_i^\eta(\|\eta\|_2/4)\}. \tag{10}$$

Finally, by Jensen's inequality, we have

$$\mathbb{E}_{X\sim\mu_X^{\text{cl}}}|f_*^{\text{poi}}(X) - f_*^{\text{cl}}(X)|^\alpha \leq \{\mathbb{E}_{X\sim\mu_X^{\text{cl}}}|f_*^{\text{poi}}(X) - f_*^{\text{cl}}(X)|\}^\alpha. \tag{11}$$

Plugging Inequalities (5), (8), (9), (10), and (11) into (1), we obtain an upper bound

$$r_n^{\text{cl}}(\widehat{f}^{\text{poi}}) \leq \frac{1}{1-\rho}r_n^{\text{poi}}(\widehat{f}^{\text{poi}}) + C\left[\lambda h_1^\eta(\|\eta\|_2/2) + \frac{\rho}{1-\rho}\max_{i=0,1}\{h_i^\eta(\|\eta\|_2/4)\}\right]^\alpha$$

$$\leq \frac{1}{1-\rho}r_n^{\text{poi}}(\widehat{f}^{\text{poi}}) + \frac{C}{(1-\rho)^\alpha}\left[\max_{i=0,1}\{h_i^\eta(\|\eta\|_2/4)\}\right]^\alpha.$$

**Upper bound of $r_n^{\text{bd}}(\widehat{f}^{\text{poi}})$.** The technique is the same. First, we decompose $r_n^{\text{bd}}(\widehat{f}^{\text{poi}})$ as

$$r_n^{\text{bd}}(\widehat{f}^{\text{poi}}) = \mathbb{E}_{\mathcal{D}_\eta^{\text{poi}}}[\mathbb{E}_{X\sim\mu_X^{\text{bd}}}\{\ell(\widehat{f}^{\text{poi}}(X), f_*^{\text{bd}}(X))\}]$$

$$\leq \mathbb{E}_{\mathcal{D}_\eta^{\text{poi}}}[\mathbb{E}_{X \sim \mu_X^{\text{bd}}}\{\ell(\widehat{f}^{\text{poi}}(X), f_*^{\text{poi}}(X))\}]$$
$$+ C\mathbb{E}_{X \sim \mu_X^{\text{bd}}}\{|f_*^{\text{poi}}(X) - f_*^{\text{bd}}(X)|^\alpha\}. \tag{12}$$

By Eq. (3), the first term

$$\mathbb{E}_{\mathcal{D}_\eta^{\text{poi}}}[\mathbb{E}_{X \sim \mu_X^{\text{bd}}}\{\ell(\widehat{f}^{\text{poi}}(X), f_*^{\text{poi}}(X))\}]$$
$$\leq \rho^{-1}\mathbb{E}_{\mathcal{D}_\eta^{\text{poi}}}[\mathbb{E}_{X \sim \mu_X^{\text{poi}}}\{\ell(\widehat{f}^{\text{poi}}(X), f_*^{\text{poi}}(X))\}] = \rho^{-1}r_n^{\text{poi}}(\widehat{f}^{\text{poi}}). \tag{13}$$

As for the second term, since $f_*^{\text{bd}}(X)$ equals zero, by Eq. (8), we have

$$\mathbb{E}_{X \sim \mu_X^{\text{bd}}}\{|f_*^{\text{poi}}(X) - f_*^{\text{bd}}(X)|\} = \mathbb{E}_{X \sim \mu_X^{\text{bd}}}\{f_*^{\text{poi}}(X)\}$$
$$= \int \frac{(1-\rho)\lambda\nu_1(x)}{\mu_X^{\text{poi}}(X=x)}\mu_X^{\text{bd}}(x)dx$$
$$= \rho^{-1}(1-\rho)\mathbb{E}_{X \sim \mu_X^{\text{cl}}}\{|f_*^{\text{poi}}(X) - f_*^{\text{cl}}(X)|\}. \tag{14}$$

Therefore, by plugging (8), (11), (13), and (14) into (12), we obtain

$$r_n^{\text{bd}}(\widehat{f}^{\text{poi}}) \leq \rho^{-1}r_n^{\text{poi}}(\widehat{f}^{\text{poi}}) + C(\frac{1-\rho}{\rho})^\alpha\left[\lambda h_1^\eta(\|\eta\|_2/2) + \frac{\rho}{1-\rho}\max_{i=0,1}\{h_i^\eta(\|\eta\|_2/4)\}\right]^\alpha$$
$$\leq \rho^{-1}r_n^{\text{poi}}(\widehat{f}^{\text{poi}}) + \rho^{-\alpha}C\left[\max_{i=0,1}\{h_i^\eta(\|\eta\|_2/4)\}\right]^\alpha,$$

which concludes the proof. $\qquad\square$

**Proof of Theorem 2**

*Proof:* **Lower bound of** $r_n^{\text{cl}}(\widehat{f}^{\text{poi}})$. By Assumption 2, we have

$$r_n^{\text{cl}}(\widehat{f}^{\text{poi}}) = \mathbb{E}_{\mathcal{D}_\eta^{\text{poi}}}[\mathbb{E}_{X \sim \mu_X^{\text{cl}}}\{\ell(\widehat{f}^{\text{poi}}(X), f_*^{\text{cl}}(X))\}]$$
$$\geq \mathbb{E}_{\mathcal{D}_\eta^{\text{poi}}}[\mathbb{E}_{X \sim \mu_X^{\text{cl}}}\{\ell(f_*^{\text{poi}}(X), f_*^{\text{cl}}(X)) - C|\widehat{f}^{\text{poi}}(X) - f_*^{\text{poi}}(X)|^\alpha\}]$$
$$\geq -C_\beta^{\alpha/\beta}C\left(\mathbb{E}_{\mathcal{D}_\eta^{\text{poi}}}[\mathbb{E}_{X \sim \mu_X^{\text{cl}}}\{\ell(\widehat{f}^{\text{poi}}(X), f_*^{\text{poi}}(X))\}]\right)^{\alpha/\beta}$$
$$+ C_\beta\mathbb{E}_{X \sim \mu_X^{\text{cl}}}\{|f_*^{\text{poi}}(X) - f_*^{\text{cl}}(X)|^\beta\}. \tag{15}$$

As for the second term, recall that $\|\eta\|_2 > 2c$ for a constant $c$. We then have

$$\mathbb{E}_{X \sim \mu_X^{\text{cl}}}|f_*^{\text{poi}}(x) - f_*^{\text{cl}}(x)| = \int \frac{\lambda\nu_1(x)}{\mu_X^{\text{cl}}(x)} \cdot \frac{\rho\mu_X^{\text{bd}}(x)}{\mu_X^{\text{poi}}(x)}\mu_X^{\text{cl}}(dx)$$
$$\geq \int_{\mathbb{R}^p\setminus S^\eta(\|\eta\|_2/2)} \frac{\lambda\nu_1(x)}{\mu_X^{\text{cl}}(x)} \cdot \frac{\rho\mu_X^{\text{bd}}(x)}{\mu_X^{\text{poi}}(x)}\mu_X^{\text{cl}}(dx)$$
$$\geq \rho\lambda\int_{\mathbb{R}^p\setminus S^\eta(\|\eta\|_2/2)} \nu_1(x)\mu_X^{\text{bd}}(x)dx$$
$$\geq \rho\lambda(1-\lambda)\min_{x \in S}\nu_1(x-\eta)\int_S \nu_1(x)dx$$
$$= \rho C_5 g_1^\eta(c), \tag{16}$$

where $S = \{x : \|x - m_i\|_2 \leq c\}$, and $C_5 = \lambda(1-\lambda)\int_S \nu_1(x)dx$ is a constant irrelevant of $\eta$. Also, since $\beta \geq 1$, by Jensen's inequality, we have

$$\mathbb{E}_{X \sim \mu_X^{\text{cl}}}\{|f_*^{\text{poi}}(X) - f_*^{\text{cl}}(X)|^\beta\} \geq [\mathbb{E}_{X \sim \mu_X^{\text{cl}}}\{|f_*^{\text{poi}}(X) - f_*^{\text{cl}}(X)|\}]^\beta. \tag{17}$$

Plugging Eqs. (5), (16), and (17) into (15), we obtain the lower bound as

$$r_n^{\text{cl}}(\widehat{f}^{\text{poi}}) \geq \rho^\beta C_5^\beta\{g_1^\eta(c)\}^\beta - C_\beta^{\alpha/\beta}C\{r_n^{\text{poi}}(\widehat{f}^{\text{poi}})\}^{\alpha/\beta}.$$

**Lower bound of $R_n^{\mathrm{bd}}$.** With a similar argument, we have

$$
\begin{aligned}
r_n^{\mathrm{bd}}(\widehat{f}^{\mathrm{poi}}) &= \mathbb{E}_{\mathcal{D}_\eta^{\mathrm{poi}}}[\mathbb{E}_{X \sim \mu_X^{\mathrm{bd}}}\{\ell(\widehat{f}^{\mathrm{poi}}(X), f_*^{\mathrm{bd}}(X))\}]\\
&\geq -C_\beta^{\alpha/\beta} C\left(\mathbb{E}_{\mathcal{D}_\eta^{\mathrm{poi}}}[\mathbb{E}_{X \sim \mu_X^{\mathrm{cl}}}\{\ell(\widehat{f}^{\mathrm{poi}}(X), f_*^{\mathrm{poi}}(X))\}]\right)^{\alpha/\beta}\\
&\quad + C_\beta \mathbb{E}_{X \sim \mu_X^{\mathrm{bd}}}\{|f_*^{\mathrm{poi}}(X) - f_*^{\mathrm{bd}}(X)|^\beta\}.
\end{aligned}
\tag{18}
$$

Plugging (13), (14), (16) and (17) into (18), we obtain

$$
r_n^{\mathrm{bd}}(\widehat{f}^{\mathrm{poi}}) \geq (1-\rho)^\beta C_5^\beta \{g_1^\eta(c)\}^\beta - C_\beta^{\alpha/\beta} C \{r_n^{\mathrm{poi}}(\widehat{f}^{\mathrm{poi}})\}^{\alpha/\beta},
$$

which concludes the proof. $\qquad\square$

**Proof of Theorem 3**

*Proof:* We prove the result for $r_n^{\mathrm{cl}}(\widehat{f}^{\mathrm{poi}})$, and the proof for $r_n^{\mathrm{bd}}(\widehat{f}^{\mathrm{poi}})$ is parallel. Without loss of generality, we assume that $\Sigma$ is a diagonal matrix with $\Sigma_{ii} = \sigma_i$, and $m_1 = 0$. Therefore,

$$
h_1^\eta(r) = h_1^\eta(|\eta^{\mathrm{T}} X| \geq r\|\eta\|_2) = 2\mathbb{P}(Z \geq r\|\eta\|_2/(\eta^{\mathrm{T}}\Sigma\eta)^{1/2}),
\tag{19}
$$

where $Z$ is a standard Gaussian random variable. Recall that $\|\eta\|_2 \geq 2c$, we have

$$
\begin{aligned}
g_1^\eta(c) &= \min_{\{\|x-\eta\|_2 \leq c\}} \nu_1(-x) = \min_{\|u\|_2 \leq \|\eta\|_2/2} (2\pi)^{-p/2}|\Sigma|^{-1/2}\exp\{-(\eta+u)^{\mathrm{T}}\Sigma^{-1}(\eta+u)\}\\
&\geq (2\pi)^{-p/2}|\Sigma|^{-1/2}\exp\{-\eta^{\mathrm{T}}\Sigma^{-1}\eta - \|\eta\|_2^2/(4\sigma_p)\},
\end{aligned}
\tag{20}
$$

where $|\Sigma|$ denotes the determinant of $\Sigma$ and the last step is due to the Cauchy inequality.

It is clear from Eq. (19) and (20) that to minimize the bounds in Theorems 1 and 2, we should choose the direction of $\eta$ to minimize $\eta^{\mathrm{T}}\Sigma^{-1}\eta$, which is exactly along the direction of $u_p$, the eigenvector of the smallest eigenvalue.

Given the direction, we next consider the magnitude of $\eta$ to achieve a successful attack. For the squared error loss, by Remark 2, we have $\alpha = 1$ and $\beta = 2$. It is also known from the Mill's inequality that the tail of a standard normal random variable $Z$ satisfies

$$
\mathbb{P}(Z \geq z) \leq \sqrt{2/\pi} z^{-1} e^{-z^2/2}, \quad \forall z > 0.
$$

Now, choosing $\eta = \|\eta\|_2 u_p$ and invoking Theorems 1 and 2, we have

$$
r_n^{\mathrm{cl}}(\widehat{f}^{\mathrm{poi}}) \lesssim r_n^{\mathrm{poi}}(\widehat{f}^{\mathrm{poi}}) + \|\eta\|_2^{-1} e^{-\eta^{\mathrm{T}}\eta/(32\sigma_p)}.
\tag{21}
$$

$$
r_n^{\mathrm{cl}}(\widehat{f}^{\mathrm{poi}}) \gtrsim e^{-\eta^{\mathrm{T}}\eta/(2\sigma_p)} - r_n^{\mathrm{poi}}(\widehat{f}^{\mathrm{poi}}).
\tag{22}
$$

A successful attack means that $r_n^{\mathrm{cl}}(\widehat{f}^{\mathrm{poi}}) \lesssim r_n^{\mathrm{cl}}(\widehat{f}^{\mathrm{cl}})$. Thus, according to Eq. (21) and Assumption 3, we only need

$$
\|\eta\|_2^{-1} e^{-\eta^{\mathrm{T}}\eta/(32\sigma_p)} \lesssim r_n^{\mathrm{cl}}(\widehat{f}^{\mathrm{cl}}) \asymp n^{-\gamma}.
$$

Taking the logarithm on both sides, the above is equivalent to

$$
\eta^{\mathrm{T}}\eta \geq C_5 \ln n,
$$

where $C_5 = 32\sigma_p\gamma$.

On the other hand, when $\eta^{\mathrm{T}}\eta \leq C_6 \ln n$, where $C_6$ is a positive constant smaller than $2\sigma_p\gamma$, we can verify that

$$
\lim_{n \to \infty} e^{-\eta^{\mathrm{T}}\eta/(2\sigma_p)}/r_n^{\mathrm{cl}}(\widehat{f}^{\mathrm{cl}}) = \infty.
$$

Therefore, Eq. (22) immediately implies that the corresponding attack is unsuccessful, and we complete the proof.

$\qquad\square$

**Proof of Theorem 4** *Proof:* The data distribution degenerates along the direction of $u$, which immediately implies that

$$h_i^u(r) = 0, \; g_i^u(r) = 0, r > 0.$$

Thus, when $\eta = s \cdot u$ for any $s > 0$, Theorem 1 gives

$$r_n^{\text{cl}}(\widehat{f}^{\text{poi}}) \lesssim r_n^{\text{poi}}(\widehat{f}^{\text{poi}}), \; r_n^{\text{bd}}(\widehat{f}^{\text{poi}}) \qquad\qquad \lesssim r_n^{\text{poi}}(\widehat{f}^{\text{poi}}). \tag{23}$$

Under Assumption 3, we know that

$$r_n^{\text{cl}}(\widehat{f}^{\text{poi}}) \gtrsim r_n^{\text{cl}}(\widehat{f}^{\text{cl}}), \; r_n^{\text{bd}}(\widehat{f}^{\text{poi}}) \gtrsim r_n^{\text{cl}}(\widehat{f}^{\text{cl}}), \tag{24}$$

which concludes the proof. □

**Proof of Theorem 5.**

*Proof:* Let $f_{*X}^{\text{cl}} = \mathbb{P}(Y \mid X)$ denote the conditional distribution with respect to the clean data distribution $\mu^{\text{cl}}$, and similarly define $f_{*X}^{\text{poi}}$ and $f_{*X}^{\text{bd}}$. Let $\widehat{f}^{\text{poi}}$ be the learned function of the conditional distributions, that is, $\widehat{f}^{\text{poi}}(X) = \widehat{f}_X^{\text{poi}}$. Analogously to the proof of Theorem 4, we have

$$
\begin{aligned}
r_n^{\text{cl}}(\widehat{f}^{\text{poi}}) &= \mathbb{E}_{\mathcal{D}_\eta^{\text{poi}}}[\mathbb{E}_{X \sim \mu_X^{\text{cl}}}\{\ell_p(\widehat{f}_X^{\text{poi}}, f_{*X}^{\text{cl}})\}] \\
&\le \mathbb{E}_{\mathcal{D}_\eta^{\text{poi}}}[\mathbb{E}_{X \sim \mu_X^{\text{cl}}}\{\ell(\widehat{f}_X^{\text{poi}}, f_{*X}^{\text{poi}}) + C\mathbb{E}_{Y \sim p}|f_{*X}^{\text{poi}}(Y) - f_{*X}^{\text{cl}}(Y)|^\alpha\}] \\
&\le \mathbb{E}_{\mathcal{D}_\eta^{\text{poi}}}[\mathbb{E}_{X \sim \mu_X^{\text{cl}}}\{\ell(\widehat{f}_X^{\text{poi}}, f_{*X}^{\text{poi}})\}] + C\mathbb{E}_{X \sim \mu_X^{\text{cl}}}\mathbb{E}_{Y \sim p}\{|f_{*X}^{\text{poi}}(Y) - f_{*X}^{\text{cl}}(Y)|^\alpha\}.
\end{aligned}
$$

The first term in the right-hand size is

$$
\begin{aligned}
\mathbb{E}_{\mathcal{D}_\eta^{\text{poi}}}[\mathbb{E}_{X \sim \mu_X^{\text{cl}}}\{\ell(\widehat{f}_X^{\text{poi}}, f_{*X}^{\text{poi}})\}] &\le (1-\rho)^{-1}\mathbb{E}_{\mathcal{D}_\eta^{\text{poi}}}[\mathbb{E}_{X \sim \mu_X^{\text{poi}}}\{\ell(\widehat{f}_X^{\text{poi}}, f_{*X}^{\text{poi}})\}] \\
&= (1-\rho)^{-1}r_n^{\text{poi}}(\widehat{f}^{\text{poi}}).
\end{aligned}
$$

The second term equals zero, because for any $x$ such that $\mu_X^{\text{cl}}(x) > 0$, we have

$$
\begin{aligned}
f_{*x}^{\text{poi}}(Y) = \mathbb{P}_{\mu^{\text{poi}}}(Y \mid X = x) &= \frac{\mathbb{P}_{\mu^{\text{poi}}}(x, Y)}{\mathbb{P}_{\mu^{\text{poi}}}(x)} \\
&= \frac{(1-\rho)\mathbb{P}_{\mu^{\text{cl}}}(x, Y)}{(1-\rho)\mathbb{P}_{\mu^{\text{cl}}}(x)} = \mathbb{P}_{\mu^{\text{cl}}}(Y \mid X = x) = f_{*x}^{\text{cl}}(Y),
\end{aligned}
$$

noting that $\mathbb{P}_{\mu^{\text{cl}}}(X + \eta) = 0$. As a result, we have

$$r_n^{\text{cl}}(\widehat{f}^{\text{poi}}) \le (1-\rho)^{-1}r_n^{\text{poi}}(\widehat{f}^{\text{poi}}) \lesssim r_n^{\text{cl}}(\widehat{f}^{\text{cl}}). \tag{25}$$

With the same argument as Inequality (25), we can obtain that

$$r_n^{\text{bd}}(\widehat{f}^{\text{poi}}) \le \rho^{-1}r_n^{\text{poi}}(\widehat{f}^{\text{poi}}) \lesssim r_n^{\text{cl}}(\widehat{f}^{\text{cl}}).$$

The above completes the proof. □

# B EXTENSIONS OF THE THREAT MODEL

## B.1 CLEAN-LABEL BACKDOOR ATTACKS

We note that the techniques used in this paper can be applied to study clean-label backdoor attacks (Barni et al., 2019; Liu et al., 2020), where the responses of poisoned data points remain unchanged. Specifically, the crux of our analysis, such as Theorem 1, involves estimating the difference between the regression function on poisoned data, $f_*^{\text{poi}}$, and that on clean data, $f_*^{\text{cl}}$ (referenced in Eq. (1), (6), and (7) in the proof of Theorem 1). Since $f_*^{\text{poi}}$ stands for the regression function with respect to the distribution of poisoned data, which is known from the attacker's perspective, we can determine $f_*^{\text{poi}}$ in the clean-label cases as well. Then, the methodology for estimating its difference from $f_*^{\text{cl}}$ and error bounds in the clean-label case would follow the same logical framework as outlined in our main paper.

| Ratio → | 0.1% | 1% | 5% | 10% | 50% | 90% | 95% | 99% |
|---|---|---|---|---|---|---|---|---|
| Clean Acc | 0.99 | 0.98 | 0.98 | 0.98 | 0.98 | 0.94 | 0.91 | 0.42 |
| Backdoor Acc | 0.41 | 0.92 | 0.98 | 0.99 | 1.0 | 1.0 | 1.0 | 1.0 |

Table 3: Effects of backdoor ratio on MNIST.

### B.2 MULTI-CLASS RESPONSES

Our framework can be extended to analyze multi-discriminators with more than 2 classes as well. One possible extension in multi-class scenarios is assuming that poisoned data have a common target label. Since $f_*^{\mathrm{poi}}$ stands for the regression function with respect to the distribution of poisoned data, which is known from the attacker's perspective, we can therefore determine $f_*^{\mathrm{poi}}$ in the multi-class cases. As a result, the difference between the regression function on poisoned data can be controlled by the difference at each coordinate respectively. The following analysis would follow the same logical framework as outlined in our main paper.

## C  FURTHER EXPERIMENTS

**Influence of poisoning ratio.** To corroborate our theoretical observation on backdoor data ratio $\rho$, we performed BadNets attack on MNIST, replacing a 2 by 2 area at the lower-right corner with pixel value 5. The results are summarized in Table 3. From the results, we find that as the poisoning ratio $\rho$ increases, the clean data accuracy is pretty stable at the beginning, and then quickly drops when $\rho$ approaches one. In the meanwhile, the backdoor data accuracy increases as $\rho$ increases, which aligns with our results at the end of Section 3.

## D  FUTHER DISCUSSION

This paper elucidates the working mechanisms of backdoor attacks and quantitatively assesses their efficacy in relation to the probabilistic distinction between backdoored and clean data. The developed insight sheds light on the success of human-imperceptible attacks. Future research includes examining the performance of backdoor attacks on vanishing backdoor data ratios $\rho$, measuring the magnitude of the backdoor trigger besides the $\ell_2$-norm, investigating backdoor attacks for sparsely pruned sub-models (Diao et al., 2023), extending the insights to distributed learning (Xian et al., 2020; Ding et al., 2022) and decentralized multimodal learning (Diao et al., 2022) scenarios.

