# OpenReview forum: "Demystifying Poisoning Backdoor Attacks from a Statistical Perspective"
_ICLR.cc/2024/Conference — ICLR 2024 poster_

### Official Review · Reviewer_76rk · 2023-10-27

**Soundness:** 3 good
**Presentation:** 3 good
**Contribution:** 2 fair
**Rating:** 6
**Confidence:** 3

**Summary:**

This paper addresses the safety risks in machine learning posed by backdoor attacks, where triggers are embedded in models to activate malicious behavior under specific conditions. It focuses on evaluating the effectiveness of backdoor attacks with constant triggers and establishing performance boundaries for models on clean and compromised data. The study explores key issues: the factors determining an attack's success, the optimal strategy for an attack, and the conditions for success with human-imperceptible triggers. Applicable to both discriminative and generative models, the findings are validated through experiments using benchmark datasets and current backdoor attack scenarios.

**Strengths:**

- This paper focuses on an important problem. It provides a fundamental understanding of the influence of backdoor attacks.
- This paper provides extensive theoretical analysis.
- This paper is easy to follow.

**Weaknesses:**

- The observation that a high poisoning ratio adversely affects the performance of clean data lacks novelty.
- The paper lacks clarity in some sections. For instance, Section 6.2.1 discusses the impact of backdoor trigger magnitudes, but fails to specify crucial details of the attack setting, such as the size of the trigger.
- The authors assert that WaNet, Adaptive Patch, and Adaptive Blend attacks are more effective than BadNets, as evidenced by a greater relative change in dimensions with low variance. However, the term "effectiveness" needs clarification. BadNets is known for its high attack success rate, so how do these methods compare under identical attack settings, including trigger size and magnitude?
- The methodology for measuring the Mean Squared Error (MSE) between clean training images and those altered by the backdoored DDPM is unclear. Given that DDPM generation is inherently a stochastic process, a more detailed explanation of this measurement technique would be beneficial.

**Questions:**

See the weakness.

---

> ### Author Response · Authors · 2023-11-16
>
> We sincerely thank the reviewer for dedicating time reviewing our work.
>
> 1. "The observation that a high poisoning ratio adversely affects the performance of clean data lacks novelty."
>
>    **Response:** Our results are non-trivial in several aspects. **First,** we offer a **quantitative** analysis, beyond mere empirical observations, about the effects of backdoor attacks. For instance, instead of the empirical observation that a high poisoning ratio can damage the model performance, we establish both lower and upper bounds for the prediction risk, as presented in Theorem 1 and 2. Those results serve as foundations for more nuanced analysis. For example, we find that a vanishing backdoor data ratio may be sufficient for successful backdoor attacks, and can even derive the minimum required amount of backdoor data in specific scenarios. **Second,** our work is pioneering in theoretically studying the performance of backdoor attacks. The closest work, to our knowledge, is Manoj & Blum (2021), which focuses on the model’s vulnerability, while ours is the first to quantify the effectiveness of these attacks. **Third,** the analytical techniques we used are also non-trivial. The main proof steps involve establishing the connection between the poisoned model and clean model, estimating the model deviation caused by backdoor data, and carefully optimizing the inequality so that the error bounds are tight in the sense of order.
>
> 2. "The paper lacks clarity in some sections. For instance, Section 6.2.1 discusses the impact of backdoor trigger magnitudes, but fails to specify crucial details of the attack setting, such as the size of the trigger."
>
>    **Response:** We present detailed specifications for the setup in Table 2 as follows. In the case of MNIST, the backdoor triggers are 2 by 2 square patches, while for CIFAR-10, 3 by 3 square patches are utilized. All backdoor triggers are positioned at the lower-right corner of the inputs, replacing the original pixels with identical values. The pixel value represents the magnitude of the backdoor trigger. The poisoning ratio is consistently set at 5% for both datasets, and ResNet models are utilized in the experiments.
>
> 3. "The authors assert that WaNet, Adaptive Patch, and Adaptive Blend attacks are more effective than BadNets, as evidenced by a greater relative change in dimensions with low variance. However, the term "effectiveness" needs clarification. BadNets is known for its high attack success rate, so how do these methods compare under identical attack settings, including trigger size and magnitude?"
>
>       **Response:** When discussing 'effectiveness,' we are referring to the consideration that backdoor attacks, exhibiting comparable clean and backdoor accuracies yet featuring less visibly discernible triggers, are deemed more effective. This is due to the fact that these subtle triggers can easily elude human eye detection, thereby enhancing their overall stealthiness. For instance, consider the WaNet/BadNets method, achieving a clean accuracy of 94.15%/94.3% and a backdoor accuracy of 99.15%/99.9% on the CIFAR-10 dataset, respectively. Despite the similar clean and backdoor accuracies of the two methods, the WaNet method is considered more effective due to its less visible trigger, as illustrated in the Figure on the official GitHub page of WaNet (https://github.com/VinAIResearch/Warping-based_Backdoor_Attack-release).
>
>
> 4. "The methodology for measuring the Mean Squared Error (MSE) between clean training images and those altered by the backdoored DDPM is unclear. Given that DDPM generation is inherently a stochastic process, a more detailed explanation of this measurement technique would be beneficial."
>
>       **Response:** The MSE is computed by comparing the averaged pixel values of the generated images with those of the target images.
>
>       The goal of backdoor attacks on diffusion models is to create images with a predefined pattern triggered by specific backdoor text input. To evaluate the effectiveness of backdoor attacks, the assessment criteria should measure the similarity or distance between the generated backdoored images and the target images. Consistent with the approach outlined in [A], we employ MSE to quantify the distances between the generated images and the target images, as an approach of evaluating the effectiveness of backdoor attacks.
>
>       Refs: [A] Chou et al, "How to backdoor diffusion models?", in CVPR 2023.
>
> We will incorporate your constructive comments and our responses into the paper. An updated version of the paper, encompassing these clarifications and extensions, will be uploaded as soon as possible.

---

> > ### Author Response · Authors · 2023-11-22
> >
> > We sincerely appreciate Reviewer 76rk for their constructive comments. In our response, we have addressed concerns regarding (1) novelty in the analysis of backdoor data ratio, (2) clarification on experiment details, (3) clarification on the term "efffectiveness", and (4) clarification on the calculation of MSE.
> >
> > As the discussion phase nears its conclusion, we kindly inquire if the reviewer has any further comments on our response. We are readily available for any additional queries they may have.
> >
> > Once again, we appreciate your time and effort in reviewing our paper.

---

> > > ### Comment · Reviewer_76rk · 2023-11-22
> > >
> > > Thanks for the author's response. While I maintain certain reservations regarding some of the conclusions and observations presented, I acknowledge that the author has addressed a number of my concerns. In light of this, I have adjusted my rating to a 6.

---

> > > > ### Author Response · Authors · 2023-11-22
> > > >
> > > > Thank you for your helpful feedback and your readiness to increase the score. We will be sure to incorporate your suggestions in our revisions.

---

### Official Review · Reviewer_bWko · 2023-10-29

**Soundness:** 3 good
**Presentation:** 3 good
**Contribution:** 4 excellent
**Rating:** 8
**Confidence:** 4

**Summary:**

From the statistical perspective, this paper theoretically analyzed the efficiency of backdoor attacks. Specifically, focusing on the binary classification and generative model, the authors relied on two assumptions to calculate the tight lower and upper boundaries of the backdoor model’s performance on the clean and poisoned test data.

**Strengths:**

Their theoretical conclusion for the efficiency of backdoor attacks matches with the empirical results. For instance, the influence of the poisoning ratio and the magnitude of the trigger signal. Moreover, they also claimed that when fixing the poisoning ratio and the magnitude of the trigger, it is more efficient to choose the trigger along the direction the density of clean data drops quickly.

**Weaknesses:**

One thing I want to mention is about the reference, as far as I know, there exist some references on the backdoor efficiency. The authors should cite them.
[1] W. Guo, B. Tondi and M. Barni, "A Temporal Chrominance Trigger for Clean-Label Backdoor Attack Against Anti-Spoof Rebroadcast Detection," in IEEE Transactions on Dependable and Secure Computing, doi: 10.1109/TDSC.2022.3233519.
[2] Yinghua Gao, Yiming Li, Linghui Zhu, Dongxian Wu, Yong Jiang, and Shu-Tao Xia. Not all samples are born equal: Towards effective clean- label backdoor attacks. Pattern Recognition, 139:109512, 2023. 2, 3
[3] Pengfei Xia, Ziqiang Li, Wei Zhang, and Bin Li. Data-efficient backdoor attacks. In Proceedings of the Thirty-First International Joint Conference on Artificial Intelligence, IJCAI-22, pages 3992–3998, 2022

**Questions:**

Is it possible to extend this theoretical framework for multi-discriminator with more than 2 classes.

---

> ### Author Response · Authors · 2023-11-16
>
> We are grateful for the reviewer's time and effort in reviewing our work.
>
> 1. "One thing I want to mention is about the reference."
>     **Response:** We sincerely thank the reviewer for pointing out those related work. All three referenced papers focus on enhancing backdoor attack efficiency by the efficiency of backdoor attacks through strategic selection of data points for poisoning. This line of research indeed offers a valuable complement to our study that considers a random poisoning strategy. In particular, Gao et al. [2] and Xia et al. [3] propose selecting data instances that significantly influence the formation of the decision boundary in the learned model, while Guo et al. [1] suggest selecting data points that lie in close proximity to the decision boundary of the clean model.
>
>   In recognition of the relevance and contribution of these works to our research context, we have included a discussion of these works in the related work section of our paper.
>
> 2. "Is it possible to extend this theoretical framework for multi-discriminator with more than 2 classes."
>
>    **Response:** Thanks for your constructive comment. The short answer is yes. For example, one possible extension in multi-class scenarios is assuming that poisoning data have a common target label. The crux of our approach involves estimating the difference between the regression function on poisoned data, $f_*^{poi}$ , and that on clean data,  $f_*^{cl}$ (referenced in Eq. (1), (6), and (7) in the proof of Theorem 1). Since $f_*^{poi}$ stands for the regression function with respect to the distribution of poisoned data, which is known from the attacker's perspective, we can determine $f_*^{poi}$ in the multi-class cases as well. Although both $f_*^{poi}$ and $f_*^{cl}$ are multi-valued functions in the multi-class cases, we can bound their difference by controlling the difference at each coordinate respectively. Then, the methodology for estimating its difference from $f_*^{cl}$ and error bounds would follow the same logical framework as outlined in our paper.
>
> We will incorporate your constructive comments and our responses into the paper. An updated version of the paper, encompassing these clarifications and extensions, will be uploaded as soon as possible.

---

> > ### Author Response · Authors · 2023-11-22
> >
> > We sincerely appreciate Reviewer bWko for their insightful and positive comments. In our response, we have addressed concerns regarding (1) related references, and (2) extension to multi-class tasks.
> >
> > As the discussion phase nears its conclusion, we kindly inquire if the reviewer has any further comments on our response. We are readily available for any additional queries they may have.
> >
> > Once again, we appreciate your time and effort in reviewing our paper.

---

### Official Review · Reviewer_DvbB · 2023-10-29

**Soundness:** 3 good
**Presentation:** 2 fair
**Contribution:** 3 good
**Rating:** 6
**Confidence:** 3

**Summary:**

This paper conducts a theoretical analysis of backdoor attacks, with a focus on addressing three key questions: (1) What are the factors that determine the effectiveness of a backdoor attack? (2) What is the optimal choice of trigger with a given magnitude? (3) What is the minimum required magnitude of the trigger for a successful attack? The paper utilizes finite-sample analysis to derive both upper and lower bounds for the success of a backdoor attack. The poisoning rate, trigger magnitude, and trigger direction are important factors influencing the success of a backdoor attack. Additionally, this paper carries out experiments on synthetic data as well as tasks involving image classification and generation. The empirical results validate the theoretical analysis.

**Strengths:**

1. The paper provides a theoretical analysis on backdoor attacks, an important topic of machine learning security.

2. A few factors that contribute to the success of a backdoor attack are studied in the paper. The choice of a trigger is particularly interesting. The insights shown in the paper can provide a theoretical guideline for further work.

3. The empirical results on synthetic data validate the theoretical analysis and also provide an explanation for generative models.

**Weaknesses:**

1. Some claims are not well validated empirically. The paper states "a large backdoor data ratio ρ will damage the performance on clean data." But there is no empirical evidence to support this claim. Also, according to the literature, a high poisoning rate usually does not significantly affect clean accuracy. It is recommended to empirically validate this claim and assess its consistency with the theories.

2. The experiment conducted in Table 2 is not clear. What does the magnitude of backdoor triggers mean? Is it the L2 norm of η, or a fixed pixel value that replaces the original pixel on the input? How large is the backdoor trigger used in this study? In addition, the formalization of backdoor trigger as η in X' = X + η is not accurate. Backdoor attacks, such as BadNets replace the original pixel values with the backdoor trigger. Otherwise, the trigger pattern is not fixed and varies on different inputs.

3. The paper seems to focus on dirty-label backdoor attacks, where the poisoned samples are assigned a target label. There is anther line of attacks that do not change the label, such as SIG [1] and reflection attack [2]. Is the proposed theoretical analysis applicable to these clean-label attacks?

[1] Barni, M., Kallas, K., & Tondi, B. (2019, September). A new backdoor attack in cnns by training set corruption without label poisoning. ICIP 2019.\
[2] Liu, Y., Ma, X., Bailey, J., and Lu, F. Reflection backdoor: A natural backdoor attack on deep neural networks. ECCV 2020.

**Questions:**

N/A

---

> ### Author Response · Authors · 2023-11-16
>
> We sincerely thank the reviewer for dedicating time reviewing our work.
>
> 1. "Some claims are not well validated empirically. The paper states "a large backdoor data ratio ρ will damage the performance on clean data." But there is no empirical evidence to support this claim. Also, according to the literature, a high poisoning rate usually does not significantly affect clean accuracy. It is recommended to empirically validate this claim and assess its consistency with the theories."
>
>    **Response:** Thanks for pointing it out. We have corrected our claim as "a large backdoor data ratio $\rho$ **could** damage the performance on clean data in some scenarios".  From our finite-sample results, a larger $\rho$  leads to the increase in both lower and upper bounds of prediction risk, thus the performance could be degraded. We have also performed additional experiments showing the influence of backdoor data ratio. Specifically, we perform BadNets attack on MNIST, replacing a 2 by 2 area at the lower-right corner with pixel value 5. The results are summarized in Table 1. As the poisoning ratio $\rho$ increases, the clean data accuracy is pretty stable at the beginning, and then quickly drops when $\rho$ approaches one.
>
>    Table 1. Effects of backdoor ratio on MNIST.
>    | Ratio →     | 0.1% | 1%   | 5%   | 10%  | 50%    | 90%  | 95%  | 99%  |
>    | ------------| ---- | ---- | ---- | ---- | ----  | ---- | ---- | ---- |
>    | Clean Acc   | 0.99 | 0.98 | 0.98 | 0.98 | 0.98  | 0.94 | 0.91 | 0.42 |
>    | Backdoor Acc | 0.41| 0.92 | 0.98 | 0.99 | 1.0  | 1.0  | 1.0  | 1.0  |
>
> 2. "The experiment conducted in Table 2 is not clear. What does the magnitude of backdoor triggers mean? Is it the L2 norm of η, or a fixed pixel value that replaces the original pixel on the input? How large is the backdoor trigger used in this study? In addition, the formalization of backdoor trigger as η in X' = X + η is not accurate. Backdoor attacks, such as BadNets replace the original pixel values with the backdoor trigger. Otherwise, the trigger pattern is not fixed and varies on different inputs."
>
>    **Response:** We present detailed specifications for the setup in Table 2 as follows. In the case of MNIST, the backdoor triggers are 2 by 2 square patches, while for CIFAR-10, 3 by 3 square patches are utilized. All backdoor triggers are positioned at the lower-right corner of the inputs, replacing the original pixels with identical values. The pixel value represents the magnitude of the backdoor trigger. The poisoning ratio is consistently set at 5% for both datasets, and ResNet models are utilized in the experiments.
>
>    This paper theoretically studies backdoor attacks with the formulation of $X^{\prime} = X + \eta$. While BadNets take the form of replacement, our formulation is consistent with the experimental setup for MNIST, where almost all images have pixel values of 0 at the lower-right corner. We will include this clarification in the paper.
>
> 3. "The paper seems to focus on dirty-label backdoor attacks, where the poisoned samples are assigned a target label. There is anther line of attacks that do not change the label, such as SIG [1] and reflection attack [2]. Is the proposed theoretical analysis applicable to these clean-label attacks?"
>
>    **Response:** Thanks very much for your insightful comment. The short answer is yes, our analytical techniques are indeed applicable to clean-label attack scenarios. Although our paper primarily focuses on dirty-label backdoor attacks, where the labels of poisoned samples are altered, the fundamental principles of our analysis can be seamlessly adapted to clean-label contexts.
>
>    Specifically, the crux of our approach involves estimating the difference between the regression function on poisoned data, $f_*^{poi}$ , and that on clean data,  $f_*^{cl}$ (referenced in Eq. (1), (6), and (7) in the proof of Theorem 1). Since $f_*^{poi}$ stands for the regression function with respect to the distribution of poisoned data, which is known from the attacker's perspective, we can determine $f_*^{poi}$ in the clean-label cases as well. Then, the methodology for estimating its difference from $f_*^{cl}$ and error bounds in the clean-label case would follow the same logical framework as outlined in our paper.
>
> We will incorporate your constructive comments and our responses into the paper. An updated version of the paper, encompassing these clarifications and extensions, will be uploaded as soon as possible.

---

> ### Author Response · Authors · 2023-11-22
>
> We sincerely appreciate Reviewer DvbB for their insightful comments. In our response, we have addressed concerns regarding (1) experiments on the influence of backdoor data ratio, (2) clarification on experiment details, and (3) extension to clean-label attacks.
>
> As the discussion phase nears its conclusion, we kindly inquire if the reviewer has any further comments on our response. We are readily available for any additional queries they may have.
>
> Once again, we appreciate your time and effort in reviewing our paper.

---

> > ### Comment · Reviewer_DvbB · 2023-11-22
> >
> > Thank authors for the response.
> >
> > I think the formulation of backdoor trigger is not quite accurate. MNIST is a very special case as it is in grey scale. But for RGB images such as CIFAR, there won't be all zero pixel values for a set of images at the same location.
> >
> > My other concerns are addressed in the response. I will keep my original rating.

---

> > > ### Author Response · Authors · 2023-11-22
> > >
> > > Thank you for your constructive feedback. We will be sure to incorporate your suggestions in our revisions.

---

### Official Review · Reviewer_DhoY · 2023-11-03

**Soundness:** 2 fair
**Presentation:** 2 fair
**Contribution:** 1 poor
**Rating:** 3
**Confidence:** 5

**Summary:**

This paper studies backdoor attacks with a constant trigger, assuming the trained classifiers are Bayesian optimal with respect to the poisoned training set.

Through this framework, they provide the following insights for backdoor attacks using a constant trigger:
1. More backdoor data can harm clean performance and can help backdoor to success.
2. Backdoor attacks can be more successful when the constant trigger has a larger magnitude.
3. Backdoor attacks can be more successful when the direction of the constant trigger points towards less popular regions (i.e. regions with smaller density).
4. Arbitrarily small backdoor data ratios may result in successful attacks.
5. If there is a direction where for all samples the corresponding support of the marginal distribution is a single point, the magnitude of the trigger can be arbitrarily small to have a successful attack.

**Strengths:**

1. Theoretical understanding of backdoor attacks is an important topics.
2. The authors demonstrate their skills in using statistical tools.

**Weaknesses:**

While I appreciate the skills demonstrated by the authors, none of the obtained insights is interesting in a sense that they are either trivial or not true without assuming the model to be Bayesian optimal with respect to the poisoned training distribution.

To be specific, insight 1&2 listed in the above Summary section are trivial (even though it may generalize to other backdoor/poison attacks); Insight 3&4&5 are trivial only when assuming the model to be Bayesian optimal but may not generalize to other (actual) learning algorithms.

To sum up, my primary concerns regarding this submission include:
1. Some key assumptions that oversimplify the problems and make the analysis probably irrelevant to practice, e.g. models are Bayesian optimal with respect to the poisoned distribution & Assumption 3 (Ordinary convergence rate) in the submission.

2. Key insights are either trivial (insight 1&2) or likely not generalizable (insight 3&4&5).


Notably, the experiments are thin but I find it acceptable for a theory paper. The major issue is not that experiments do not provide enough supports. The issue is that there is not really much insights worth supporting.

**Questions:**

Please see the weakness section above.

---

> ### Author Response · Authors · 2023-11-16
>
> We are grateful for the reviewer's time and effort in reviewing our work. However, we think there are some misunderstandings in reviewer's comments and would like to clarify them.
>
> 1. "Some key assumptions that oversimplify the problems and make the analysis probably irrelevant to practice, e.g. models are Bayesian optimal with respect to the poisoned distribution & Assumption 3 (Ordinary convergence rate) in the submission."
>
>    **Response:** **First,** contrary to the reviewer's understanding, our work does not assume that models are Bayesian optimal. We focus on analyzing the prediction error of the poisoned model $\hat{f}^{poi}$, which is evaluated as the difference between $\hat{f}^{poi}$ and the unknown ground truth $f_*$. We kindly request the reviewer to reconsider our setup and theoretical results in this light. **Second,** our finite-sample analysis establishes error bounds on backdoor model performance without relying on Assumption 3. Our results quantitatively measure the impact of critical factors in backdoor attacks, such as data poisoning ratio and trigger direction. **Third,** under Assumption 3, we derive the rate of the poisoned model's risk in the asymptotic regime. This serves as a practical example of how our results can be applied for further analysis. Remark 4 clarifies that Assumption 3 is generally valid for commonly used function classes, such as the Holder class. **Fourth,** in practical applications where Assumption 3 cannot be verified or does not hold, we have conducted experiments to corroborate the developed theory and insights. Specifically, we performed state-of-the-art attacks (BadNets, WaNet, Adaptive Patch, and Adaptive Blend) on both MNIST and CIFAR10 datasets in Section 6.2.1, and the results align with our insights such as the attack is more successful when the trigger causes poisoned data and clean data are more separated in the sense of probability.
>
>
> 2.   "Key insights are either trivial (insight 1&2) or likely not generalizable (insight 3&4&5)."
>
>      **Response:** Our insights are non-trivial in several aspects. **First,** we offer a **quantitative** analysis, beyond mere empirical observations, about the effects of backdoor attacks. For instance, instead of empirical observations such that a trigger larger in magnitude has a higher attack success rate, Theorem 3 specifies the minimum trigger length (i.e., $\ln(n)$ ) necessary for a successful attack in Gaussian distribution scenarios. **Second,** our work is pioneering in theoretically studying the performance of backdoor attacks. The closest work, to our knowledge, is Manoj & Blum (2021), which focuses on the model’s vulnerability, while ours is the first to quantify the effectiveness of these attacks. **Third,** the analytical techniques we used are also non-trivial. The main proof steps involve establishing the connection between the poisoned model and clean model, estimating the model deviation caused by backdoor data, and carefully optimizing the inequality so that the error bounds are tight in the sense of order.
>
>      The generalizability of our work are demonstrated in two aspects. **First,** we perform finite-sample analysis and provide tight error bounds with minimal assumptions, only requiring the loss function is Holder continuous and the input distribution is mostly concentrated in a bounded area. Those results can be the foundation for further analysis under various scenarios, including the asymptotic scenario discussed in our work.  **Second,** for practical applications where our technical assumptions may not be directly verifiable, we conducted experiments using both MNIST and CIFAR10 datasets and SOTA backdoor attacks (BadNets, WaNet, Adaptive Patch, and Adaptive Blend), and the results corroborate our theoretical findings.
>
> We hope these clarifications address the concerns raised and further illuminate the significance and applicability of our work.

---

> > ### Comment · Reviewer_DhoY · 2023-11-19
> >
> > If I understand correctly, a critical and specific claim in this rebuttal is that, and I quote, 'our work does not assume that models are Bayesian optimal'.
> >
> > **This is NOT true.** While I expect the authors to be more familiar with their own proofs, it is possible that the authors did not even realize they were making such an assumption during their analysis. Thus I will explain to you in details when you made this assumption for one of your results (actually), **as an example**.
> >
> > For example, for the inequality in your Theorem 1,
> > $r_n^{cl}(\hat{f}^{poi}) \leq \frac{1}{1-\rho} r_n^{poi}(\hat{f}^{poi}) + \frac{C}{(1-\rho)^\alpha}\left[\max_{i=0,1}\\{h_i^\eta(||\eta||_2 / 4)\\}\right]^\alpha$, according to Appendix A:
> > >The first term $\frac{1}{1-\rho} r_n^{poi}(\hat{f}^{poi})$
> > is an upper bound of $E_{D_{\eta}^{poi}}[E_{X\sim\mu_X^{cl}} \\{\ell(\hat{f}^{poi}(X),  f_*^{poi}(X))\\}]$, which is comparing the poisoned classifier \hat{f}^{poi} and the **Bayesian classifier on the poisoned data** $f_*^{poi}$
> >
> > >The second term is an upper bound of
> > $C\cdot E_{X \sim \mu_X^{cl}}\\{|f_*^{poi}(X) - f_*^{cl}(X)|^\alpha\\}$, which is comparing the **Bayesian classifier on the poisoned data** $f_*^{poi}$ to the clean Bayesian classifier $f_*^{cl}$.
> >
> > However, in Section 3, when deriving insights from this inequality, **the first term is ignored by claiming that 'For many classical learning algorithms, this statistical risk vanishes as the sample size goes to infinity' and only the second term is used in the analysis**. Thus the assumption is made here and your insights corresponding to this part are derived through assuming **Bayesian classifier with respect to the poisoned data**.
> >
> > Feel free to ask if this is not clear for you.

---

> > > ### Author Response · Authors · 2023-11-19
> > >
> > > We sincerely thank the reviewer's detailed explanation of their concerns. We believe the reviewer has some misunderstanding regarding our work. To summarize our responses: 1. The use of a Bayes-optimal learning algorithm is **NOT required** for the proof of our theorems. 2. We mention the case of vanishing statistical risk (common for many widely-used learning algorithms) as an illustrative example to demonstrate the potential for further analysis based on our theorems.
> > >
> > > **First**, as outlined in our paper's Section 2, $f_*^{cl}$ and $f_*^{poi}$ are the **underlying regression functions** under clean and poisoned data distributions respectively, and $\hat{f}$ represents the estimator of them. **It's important to note that the concepts like Bayes error and Bayes classifier are specific to classification rules.** For example, $g=1_{\{f_*^{cl}\geq 0.5\}}$ is a Bayes classifier, where $1_{\{\cdot\}}$ is an indicator function, while $f_*^{cl}$ itself is not a classifier. For further clarity on the definition of Bayes error and Bayes optimal rule, please see, e.g., [1, 2].
> > >
> > > [1] *Devroye L, Györfi L, Lugosi G. A probabilistic theory of pattern recognition[M]. Springer Science & Business Media, 2013. Chapter 2.*
> > >
> > > [2]  *Lin Y. A note on margin-based loss functions in classification[J]. Statistics & probability letters, 2004, 68(1): 73-82.*
> > >
> > > **Second**, our proofs and statements of theorems are **independent of the estimator's optimality**. In Theorem 1, which the reviewer highlighted, we explicitly state that the upper bounds include two terms. The first term involves the ordinary statistical risk $r_n^{poi}(\hat{f}^{poi})$, and the second term is associated with the backdoor trigger $\eta$. To this point, we do not pose any constraints on the optimality of estimator $\hat{f}$.
> > >
> > > **Third**, the reviewer commented that "the first term is ignored when deriving insights". We consider some concrete scenarios **when explaining the potential implications of Theorem 1.** One such scenario is where the statistical risk approaches zero.  As mentioned in our first point, since $\hat{f}$ is a regression function rather than a classifier, $\hat{f}$  has a vanishing statistical risk is a different thing from it is Bayes optimal. Again, the proofs of theorems do not require $\hat{f}$ to be Bayes optimal.
> > >
> > > **Fourth**, vanishing statistical risk, as known as consistency, **is a common property for a various learning algorithms**, such as k-nearest neighbors, kernel estimates, and neural networks (see, e.g., [3]).
> > >
> > > [3] Györfi L, Kohler M, Krzyzak A, et al. A distribution-free theory of nonparametric regression[M]. New York: Springer, 2002.
> > >
> > > We hope that these clarifications adequately address your queries and look forward to further discussions.

---

> > > > ### Comment · Reviewer_DhoY · 2023-11-21
> > > >
> > > > I want to thank the authors for their responses.
> > > >
> > > > While I agree it is not rigorous to simply use 'Bayesian optimal classifier' to describe your assumption, I believe the authors now understand my primary concerns based on their latest response and how it is not addressed.
> > > >
> > > > ---
> > > >
> > > > Here are my comments:
> > > >
> > > > Again, using Theorem 1 as an example. Yes, the proof of Theorem 1 does not rely on that Bayesian assumption of models, or as you would like to call it, vanishing statistical risk (with respect to a Bayesian model). **However**, after you ignore the first term in analyzing Theorem 1, any further insights you have are built on the assumption. In another word, the insights only apply to $f_*^{poi}(x) = \frac{P_{\mu^{poi}}(Y=1, X=x)}{\mu_X^{poi}(X=x)}$ but not arbitrary models. At least for now, I do not find it well-motivated to claim that the insights from analyzing $f_*^{poi}$ are significant.
> > > >
> > > > I would suggest authors to properly include their assumptions when highlighting their theoretical results.
> > > > To summarize, while I totally believe the authors can be very skillful, I have major concerns regarding the significance of the insights (theoretical implications), especially given the primary assumptions.

---

> ### Author Response · Authors · 2023-11-21
>
> We appreciate the reviewer's time in reviewing our responses. We would like to clarify the misunderstanding in reviewer's comment saying that our results "only apply to $f_*^{poi}$ but not arbitrary models."
>
> **First**, as acknowledged by the reviewer, our theorems are independent of the learning algorithms and thus the estimator (i.e., poisoned model) $\hat{f}^{poi}$. We contextualized $\hat{f}^{poi}$ in some concrete scenarios **to elucidate the potential implications of Theorem 1.**
>
> **Second**, the vanishing statistical risk, or consistency, meaning that $r_n^{poi}(\hat{f}^{poi}) = E\\{\ell(\hat{f}^{poi},f_*^{poi})\\}$ will converge to zero as the sample size $n$ goes to infinity. Here, $f_*^{poi}$ denotes the underlying regression function with respect to the poisoned data distribution, **while $\hat{f}^{poi}$ is the estimator we are analyzing**. In other words, the concept of consistency is specific to $\hat{f}^{poi}$, and our insights are pertinent to any consistent estimator $\hat{f}^{poi}$. As mentioned in the fourth point in the previous responses, **this consistency property is common for $\hat{f}^{poi}$ obtained from various learning algorithms**, such as k-nearest neighbors, kernel estimates, and neural networks (see, e.g., [3]). Notably, most existing literature on backdoor attacks presupposes the use of neural networks as the learning algorithm, as seen in, e.g., [4,5]. As a result, our insights implied by Theorem 1 are applicable to most scenarios encountered in practice.
>
> [3] Györfi L, Kohler M, Krzyzak A, et al. A distribution-free theory of nonparametric regression[M]. New York: Springer, 2002.
>
> [4] Li Y, Jiang Y, Li Z, et al. Backdoor learning: A survey[J]. IEEE Transactions on Neural Networks and Learning Systems, 2022.
>
> [5] Gao Y, Doan B G, Zhang Z, et al. Backdoor attacks and countermeasures on deep learning: A comprehensive review[J]. arXiv preprint arXiv:2007.10760, 2020.
>
> We hope that these clarifications adequately address your concerns and look forward to continued dialogue.

---

> ### Comment · Reviewer_DhoY · 2023-11-21
>
> I think one major disagreement here is whether we can consider this specific definition of 'vanishing statistical risk' used in this work as a weak assumption.
>
> Intuitively, I am conservative about such an assumption, especially in the context of deep learning and in the context of data poisoning and/or backdoor attacks. Here is a high-level explanation: In general, the behavior of learned models depend not only on data, but also on the prior knowledge encoded in the learning algorithms. For example, the design of convolutions in vision tasks incorporates the prior knowledge of translation invariance; When fine-tuning a pre-trained model, we are essentially using that pre-trained model as part of prior. In the context of poisoning/backdoor attacks, there are many filtering/detection based defenses and filtering/detection of poisoned/backdoor samples can never be possible without assuming some priors. Thus it looks logically problematic to claim that the 'vanishing statistical risk' defined in this paper is a weak assumption, since it is intuitively a 'distance' measure between a model and the $f_*^{poi}(x) = \frac{P_{\mu^{poi}}(Y=1, X=x)}{\mu_X^{poi}(X=x)}$, which obviously fail to incorporate the effect of different priors.
>
> It seems that authors are referring to [3] to support that the claimed 'vanishing statistical risk' is generally true for neural networks. It would help if the authors can (1) point out which part of [3] provides a proof for this; and (2) specify formally what is the theoretical assumptions for this claim (i.e. in what conditions this is true).
>
> I apologize to authors in advance since it is likely that I won't be able to reply very frequently in the following a day or two. But I will make sure to review any information you provide and help to reach a well-informed decision along with other reviewers.
>
> [3] Györfi L, Kohler M, Krzyzak A, et al. A distribution-free theory of nonparametric regression[M]. New York: Springer, 2002.

---

> ### Author Response · Authors · 2023-11-22
>
> We thank the reviewer for actively participating in the discussion. Your question on why a model does not incorporate priors can work well is actually an excellent illustration of the importance of statistical learning theory in understanding practical applications, highlighting the meaning and contribution of our work. In response, we focus on the following facts.
>
> 1. **Consistency is a basic requirement for a "good" estimator, regardless of the underlying data distribution, or priors as you mentioned.** To see it, we quote the comments below from a highly-cited non-parametric regression book [3]:
>
>     > The first and weakest property an estimate should have is that, as the
>     > sample size grows, it should converge to the estimated quantity, i.e., the error of the estimate should converge to zero for a sample size tending to infinity.  [3, Section 1.6]
>
>     Specifically, let $m(x)=E(Y|X)$ for a distribution of $(X,Y)$ and $\{m_n(x)\}$ be a sequence of estimators given sample size $n$. With squared error loss, $m_n(x)$ is called weakly universally consistent [3, Section 1.6] if $\lim_{n\to\infty} E\{(m_n(X) - m(X))^2\} = 0$ **for all distributions of $(X,Y)$ with $E(Y^2)<\infty$**. Thus, a weakly universally consistent estimate $\hat{f}^{poi}_n$ implies "vanishing statistical risk."
>
> 2. **Prevalence and proofs of universally consistent estimates in literature.** In particular, [3] gives proofs for k-nearest neighbors in Section 6.2, kernel estimates in Section 5.2, and two-layer neural networks in Section 16.2. More results on the consistency of deep neural networks, such as convolutional neural networks, are presented in works like [6, Section 17.1] and [7]. For rigorous derivations, we refer the reviewer to those works due to their technical complexities.
>
>     [3] Györfi L, Kohler M, Krzyzak A, et al. A distribution-free theory of nonparametric regression[M]. New York: Springer, 2002.
>
>     [6] Anthony M, Bartlett P L, Bartlett P L. Neural network learning: Theoretical foundations[M]. Cambridge: Cambridge Univ. Press, 2009.
>
>     [7] Lin S B, Wang K, Wang Y, et al. Universal consistency of deep convolutional neural networks[J]. IEEE Transactions on Information Theory, 2022, 68(7): 4610-4617.

---

### Meta-Review · Area_Chair_vWue · 2023-12-10

**Metareview:**

This work presented theoretic analysis about the critical factors to the backdoor effectiveness based on statistical tools, and provided 5 observations.

There are detailed and constructive comments from reviews, as well as  adequate discussions between reviewers and authors.
While most reviews recognized the value of theoretical analysis of backdoor learning, there are some important concerns about the non-triviality of the first two observations and the practicality of required assumptions of other observations. The authors attempted to provide some explanations about above concerns, but two reviewers still keep the reservation attitude about some claims/observations.

I throughly read the paper, all reviews and discussions. My opinion is that the first two observations are indeed trivial, although the analysis tools are non trivial and some bounds are provided; in terms of analysis about another three observations, the required assumptions and the assumptions’s practicality and scope must be clearly demonstrated to avoid over-claim and misguiding the research community.

Finally, considering the rarity and value of theoretical analysis in backdoor learning, which may inspire more future studies, I would like to recommend accept of this submission. But, it is required to adopt the objective and cautious tone about the observations, with clear demonstration of the required assumptions and the scope, to avoid misguide the research community.

**Justification For Why Not Higher Score:**

There are still concerns about the novelty of presented observations and the practicality of the adopted assumptions.

**Justification For Why Not Lower Score:**

Considering the rarity and value of theoretical analysis in backdoor learning, which may inspire more future studies, I would like to recommend accept of this submission.

---

### Decision · Program_Chairs · 2024-01-16

Accept (poster)